# On Locality of Local Explanation Models

**Sahra Ghalebikesabi**[*]
University of Oxford
sahra.ghalebikesabi@stats.ox.ac.uk

**Lucile Ter-Minassian**[*]
University of Oxford
lucile.ter-minassian@stats.ox.ac.uk

**Karla Diaz-Ordaz**
The London School of Hygiene & Tropical Medicine & The Alan Turing Institute

**Chris Holmes**
University of Oxford & The Alan Turing Institute

## Abstract

Shapley values provide model agnostic feature attributions for model outcome at a particular instance by simulating feature absence under a global population distribution. The use of a global population can lead to potentially misleading results when local model behaviour is of interest. Hence we consider the formulation of neighbourhood reference distributions that improve the local interpretability of Shapley values. By doing so, we find that the Nadaraya-Watson estimator, a well-studied kernel regressor, can be expressed as a self-normalised importance sampling estimator. Empirically, we observe that Neighbourhood Shapley values identify meaningful sparse feature relevance attributions that provide insight into local model behaviour, complimenting conventional Shapley analysis. They also increase on-manifold explainability and robustness to the construction of adversarial classifiers.

## 1 Introduction

The ability to correctly interpret a prediction model is increasingly important as we move to widespread adoption of machine learning methods, in particular within safety critical domains such as health care [22, 17]. In this paper, we consider the task of attributing the features $\{1, \ldots, m\}$ of a complex machine learning model $f : \mathbb{R}^m \to \mathbb{R}^l$, abstracted as a function that predicts a response given a test instance $x \in \mathbb{R}^m$, given only black-box access to the model. We especially focus on two popular model-agnostic feature removal based local explanation models, namely LIME [32] and SHAP [26]. However our findings are applicable to other local explanation models that we do not consider in this paper. As these methods are often described as fitting a local surrogate model to the black box [34], a natural question is: how 'local' are local explanation methods?

As a simple motivating example as to why this question matters, consider a black box model given by $f(x) = \mathbb{I}(x_1 > 0)2x_2^2 - \mathbb{I}(x_1 \leq 0)x_2^2$ where $\mathbb{I}(\cdot)$ denotes the indicator function. When attributing the local feature importance at a test instance $x = (x_1, 2)$, with $x_2$ fixed at 2, we would expect Feature-1 to receive a higher absolute attribution when $x$ is closer to the decision boundary at $x_1 = 0$. In Figure 1 we report the results on this example from LIME and SHAP as well as for our proposed 'Neighbourhood SHAP' approach. We observe that Neighbourhood SHAP assigns Feature-1 a smaller attribution, the higher the absolute value of $x_1$ is. SHAP and LIME, however, assign Feature-1 an attribution which is constant either side of $x_1 = 0$ which illustrates that these methods capture global

---

[*]equal contribution

35th Conference on Neural Information Processing Systems (NeurIPS 2021).

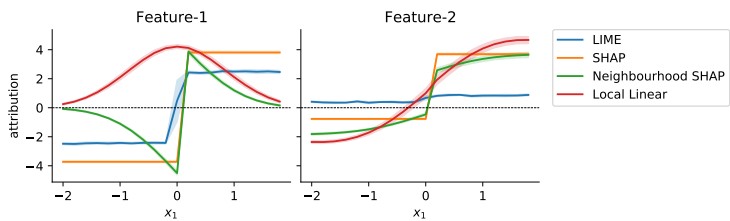

Figure 1: Attributions at $x = (x_1, 2)$ with $x_1$
varying for a reference distribution of $X \sim \text{Normal}(0, 1)$
and black box $f(x) = \mathbb{I}(x_1 > 0)2x_2^2 - \mathbb{I}(x_1 \leq 0)x_2^2$.

model behaviour. The figure also shows that training a local linear approximation to the black box [30, 8] is misleading since Feature-2 receives a significantly positive attribution for $x_1 \in [-0.4, 0]$, even though Feature-2 contributes clearly negatively to the model outcome whenever $x_1 < 0$.

This motivates the following contributions

1. We propose Neighbourhood SHAP (Section 3) which considers local reference populations for prediction points as a complimentary approach to SHAP. By doing so, we show that the Nadaraya-Watson estimator at $x$ can be interpreted as an importance sampling estimator where the expectation is taken over the proposed neighbourhood. Empirically, we find that greater locality increases the number of model evaluations on the data manifold and with this the robustness of the attributions against adversarial attacks.

2. We consider how smoothing can also be used to stabilise SHAP values (Section 4). We quantify the loss in information incurred by our smoothing procedure and characterise its Lipschitz continuity.

## 2 Background

We begin with a short introduction to Shapley values – the quantity of interest of the SHAP optimisation procedure. For a pre-defined value function $v(T, x)$ that takes a set of features $T \subseteq \{1, ..., m\}$ as input, the Shapley value $\phi_v(j, x)$ of feature $j$ measures the expected change in the value function from including feature $j$ into a random subset of features $S \subseteq \{1, ..., m\} \setminus \{j\}$ (without $j$)

$$\phi_v(j, x) = \mathop{\mathbb{E}}_{p(S)} \left[ v(S \cup \{j\}, x) - v(S, x) \right]$$

where the expectation is taken over the feature coalitions whose distribution is defined by $P(S) = \frac{|S|!(m-|S|-1)!}{m!}$. This choice of probability distribution ensures that sampling a set of size $k$ has the same probability as sampling one of size $l$, $P(\{S \mid |S| = k\}) = P(\{S \mid |S| = l\})$ for $k, l \in \{0, ..., m-1\}$.

The choice of value function for explanation-based modelling of feature attributions at an instance $x$ has been the subject of recent debates [1, 25, 27]. The consensus is to take the expectation of the black box algorithm $f$ at observation $x$ over the not-included features $\overline{S}$ using a reference distribution $r(X_{\overline{S}}^* \mid x)$ such that

$$v(S, x) = \mathop{\mathbb{E}}_{r(X_{\overline{S}}^* \mid x)} [f(x_S, X_{\overline{S}}^*)]$$

for $\overline{S} := \{1, \ldots, m\}/S$ and the operation $(x_S, x_{\overline{S}})$ denoting the concatenation of its two arguments. Marginal Shapley values [26, 25] define $r(X_{\overline{S}}^* \mid x) := p(X_{\overline{S}}^*)$ where $p$ denotes the marginal data distribution. Conditional Shapley values [1] set the reference distribution equal to the conditional distribution given $x_S$, $r(X_{\overline{S}}^* \mid x) := p(X_{\overline{S}}^* | X_S^* = x_S)$. All in all, the Shapley value $\phi(j, x)$ is characterised by the expected change in model output, comparing the output when we include $j$ in the model, i.e. integrate out some randomly sampled features $\overline{S} \setminus \{j\}$, with the model output where feature $j$ is not included, i.e. we integrated out some randomly sampled features including $j$, $\overline{S}$

$$\phi(j,x) = \mathbb{E}_{p(S)}\left[\mathbb{E}_{r(X^*_{\overline{S}\setminus\{j\}}\mid x)}[f(x_{S\cup\{j\}}, X^*_{\overline{S}\setminus\{j\}})] - \mathbb{E}_{r(X^*_{\overline{S}}\mid x)}[f(x_S, X^*_{\overline{S}})]\right].$$

As we see, Shapley values are computed by estimating the change in model outcome when some features are integrated out over the reference distribution $r(X^*_{\overline{S}}\mid x)$, which has so far been defined as either the marginal or conditional *global* population. For marginal Shapley values, the interpretation simplifies: The Shapley value of feature $j$ is the expected change in model outcome when we sample a random individual $x^*$ from the global statistical population and set its feature $j$ equal to $x_j$ (after we already set a random set of features $S \in \{1, \ldots, m\} \setminus \{j\}$ equal to $x_S$). This motivates our proposal in Section 3 of neighbourhood distributions where we instead sample a random individual from the immediate neighbourhood of $x$, as outlined in the next section.

Computing Shapely values is challenging in high-dimensional feature spaces, which motivates the widely adopted KernelSHAP approach [26] that estimates the Shapley values of all features by empirical risk minimisation of

$$\mathbb{E}_{p(S)}[(\mathbb{E}_{r(X^*_{\overline{S}}\mid x)}[f(x_S, X^*_{\overline{S}})] - g(S))^2] \approx \sum_{l=1}^{L}\sum_{i=1}^{C} w_i(f(x_{S_i}, x^*_{l,\overline{S}_i}) - g(S_i))^2 \tag{1}$$

where $g(S) = \phi_0 + \sum_{j=1}^{m}\phi(j,x)\mathbb{I}(j \in S)$ is a linear explanation model with the Shapley values as its coefficients, $\{x^*_{l,\overline{S}_i}\}_{l=1}^{L}$ are i.i.d. $L$ draws from the respective global reference distributions, $\{S_i\}_{i=1}^{C}$ is a set of size $C$ of sampled coalitions, and the weights $w_i$ are defined by the KernelSHAP weights [26]. LIME optimises a similar generalised expectation – also sampling references from a global distribution. To improve local fidelity of Tabular LIME, [32] propose to define the weights as $w_i = \exp(-(|S_i|-m)^2/\sigma^2)$ for a bandwidth $\sigma$. While this weighting increases the importance of $f(x_{S_i}, x^*_{l,\overline{S}_i})$ proportional to the size of $S$, it however does not ensure that higher weights are assigned to model evaluations for observations closer to $x$.

A simple solution to the locality problem is to fit a local linear approximation in the form of a tangent line that predicts the black box in a small neighbourhood around $x$, as in [30, 8, 41, 43]. Such an approach has however several drawbacks compared to SHAP (and thus Neighbourhood SHAP) such as higher instability, less interpretability, and assuming a fixed parametric form. While SHAP (and Neighbourhood SHAP) does not make any assumptions on the form of $f$ in the feature space, local linear approximations assume linearity of the black box in a neighbourhood. As a consequence, this may result in misleading attributions, as was demonstrated in Figure 1. See Supplement A for a detailed discussion of local approximating models versus local reference populations.

## 3 Neighbourhood SHAP

Shapley values – similarly to other feature removal methods – employ a global reference distribution when computing attributions. This can lead to surprising artefacts as illustrated in Figure 2. To increase the local fidelity of Shapley values, we propose to sample from a well-defined local reference population instead. Having selected a distance metric $D$, such as the Euclidean distance or the more powerful Random Forests [7], we define a distance-based distribution $d : \mathbb{R}^m \to \mathbb{R}$ that is centred around $x$, such as the exponential kernel $d(x^*_{\overline{S}}\mid x_{\overline{S}}) = \exp(-D(x_{\overline{S}}, x^*_{\overline{S}})^2/\sigma^2_{nbrd})$. Further, we define the local neighbourhood distribution as $n(x^*_{\overline{S}}\mid x) = n_c \cdot d(x^*_{\overline{S}}\mid x_{\overline{S}}) \cdot r(x^*_{\overline{S}}\mid x)$ where $r(x^*_{\overline{S}}\mid x)$ can be any marginal or conditional reference distribution and $n_c$ is the normalising constant. This choice ensures that we sample neighbourhood values not only considering the metric space w.r.t. $x$ but also the data distribution. This leads to a proposed change to the optimisation problem of eq. (1) to the following Neighbourhood SHAP minimisation

$$\mathbb{E}_{p(S)}\left[\left(\mathbb{E}_{n_c d(X^*_{\overline{S}}\mid x_{\overline{S}})r(X^*_{\overline{S}}\mid x)}[f(x_S, X^*_{\overline{S}})] - g(S)\right)^2\right].$$

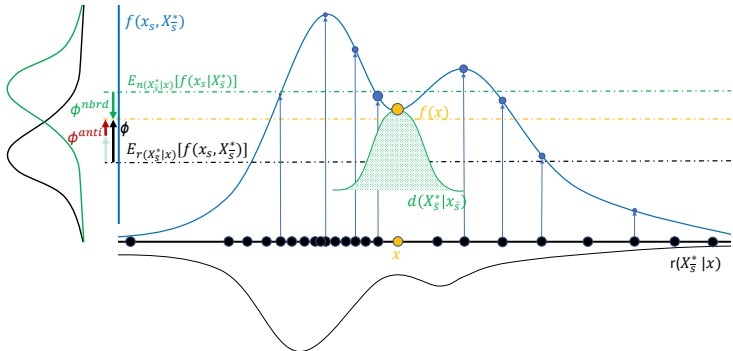

Figure 2: When sampling $\{x_i^*\}_{i=1}^L$ (black dots) from reference distribution $r(X_{\overline{S}}^* \mid x)$ (here $S = \emptyset$), the Shapley value $\phi$ at $x$ is positive since $f(x)$ is larger than $\mathbb{E}_{r(X_{\overline{S}}^* \mid x)}[f(x_S, X_{\overline{S}}^*)]$. In contrast, Neighbourhood SHAP $\phi^{nbrd}$ is negative since $\mathbb{E}_{n(X_{\overline{S}}^* \mid x)}[f(x_S, X_{\overline{S}}^*)]$ is larger than $f(x)$. This difference results from the fact that, first, the model outcome has a local minimum at $x$, and second, $f(x_S, X_{\overline{S}}^*)$ takes its smallest values at the tails of the data distribution (right-skewed density of $f(x_S, X_{\overline{S}}^*)$ when $X_{\overline{S}}^* \sim p(X_{\overline{S}}^*)$, black line on the left). SHAP only captures that $f(x)$ is higher than the average model outcome but not that $f(\cdot)$ is smaller at $x$ than it is for any other close observation – this is reflected by Neighbourhood SHAP.

Instead of estimating the neighbourhood distribution, we approximate the expectation of the model outcome in the neighbourhood around $x$ using self-normalised importance sampling [13] with proposal distribution $r(X_{\overline{S}}^* \mid x)$

$$\mathbb{E}_{n(X_{\overline{S}}^* \mid x)}[f(x_S, X_{\overline{S}}^*)] = \mathbb{E}_{r(X_{\overline{S}}^* \mid x)}\left[n_c \cdot d(X_{\overline{S}}^* \mid x_{\overline{S}})f(x_S, X_{\overline{S}}^*)\right] \approx \frac{\sum_{i=1}^L d(x_{i,\overline{S}}^* \mid x_{\overline{S}})f(x_S, x_{i,\overline{S}}^*)}{\sum_{i=1}^L d(x_{i,\overline{S}}^* \mid x_{\overline{S}})}.$$

While our proposal, Neighbourhood SHAP, weights the $f(x_S, x_{i,\overline{S}}^*)$ based on a distance metric to $x$, KernelSHAP uses uniform weights, i.e. $d(x_{\overline{S}}^* \mid x_{\overline{S}}) = 1$. We note that the proposed local neighbourhood sampling scheme has a convenient form which corresponds to the well-known Nadaraya-Watson estimator [28, 42, 35] used for kernel regression. Kernel regression is a non-parametric technique to model the non-linear relationship between a dependent variable $Y$ (here, $f(x_S, X_{\overline{S}}^*)$) and an independent variable $Z$ (here, $X_{\overline{S}}^*$), by approximating the conditional expectation $\mathbb{E}[Y \mid Z]$ (here, $\mathbb{E}_{r(X_{\overline{S}}^* \mid x)}[f(x_S, X_{\overline{S}}^*) \mid X_{\overline{S}}^*]$). While the form of the Nadaraya-Watson estimator has so far been justified from a kernel theory perspective (Supplement F), we show that it can be interpreted as an importance sampling estimator.

**Proposition 1.** *The Nadaraya-Watson estimator* $\mathbb{E}\widehat{\mathbb{E}}[Y \mid Z = z^*] = \frac{\sum_{i=1}^L d(z_i \mid z^*)y_i}{\sum_{j=1}^L d(z_j \mid z^*)}$, *where* $d(z \mid z^*)$ *is a kernel function, is a consistent self-normalised importance sampling estimator of* $Y(Z)$ *with proposal distribution* $p(z)$ *and desired distribution proportional to* $p(z)d(z \mid z^*)$.

As pointed out in Supplement B, all Shapley axioms [26, 39] still hold true for the Neighbourhood SHAP. Now, by linearity, we can quantify the difference between SHAP and Neighbourhood SHAP as 'Anti-Neighbourhood SHAP' (see Supplement D). Looking at this difference might be of value to characterise the information loss when contrasting an instance to the global population instead of to a local neighbourhood. Finally, we also derive a variance estimator of Shapley values computed with the Shapley formula in Supplement J.

**On-Manifold Explainability.** A major disadvantage of marginal Shapley values and LIME is that the concatenated data vectors $(x_S, x_{i,\overline{S}}^*)$ for a sampled reference $x_i^*$ do not necessarily lie on the data manifold [15, 10]. This has two serious ramifications: 1) the model is evaluated in regions that lie off the data manifold where it might behave unexpectedly, and be unrepresentative for the data population; a similar problem was described by [24] who note that removal based methods induce

bias in model explanations if removal is modelled by imputing with observations that are far from the actual test instance; and 2) adversaries can use an out-of-distribution (OOD) classifier trained to distinguish real-data from simulated concatenated data and, through this, construct a model whose Shapley values look fair even though the model is demonstrably unfair on the real-data domain [37]. To circumvent this problem, [16, 1, 11] propose the use of conditional instead of marginal reference distributions. However, using conditional reference distributions changes the interpretability of Shapley values – i.e. unrelated features get a non-zero attribution – and thus, their use is controversial [25]. A marginal Neighbourhood SHAP approach in contrast can achieve on-manifold explainability while keeping the properties of marginal Shapley values for small enough $\sigma_{nbrd}$ if the data manifold is to some extent coherent (see Figure 3).

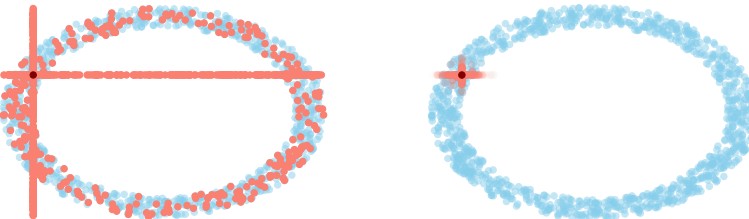

Figure 3: Concatenated data (pink dots) used for model evaluations for the computation of KernelSHAP (left) and Neighbourhood SHAP ($\sigma_{nbrd} = 0.1$, right) at a randomly sampled instance (maroon dots) where the data manifold is a ring in $\mathbb{R}^2$. Even though the background references (blue dots) lie on the data manifold, marginal Shapley values are evaluated at instances that lie off the data manifold.

**Choice of Bandwidth.** For $\sigma_{nbrd} \to \infty$, Neighbourhood SHAP will be equal to KernelSHAP, while it converges to 0 for $\sigma_{nbrd} \to 0$. Small neighbourhoods thus induce regularisation in the predictions which we also observe empirically in Section 5. While SHAP values add up to $\sum_{j=1}^{m} \phi^{nbrd}(j, x) = f(x) - \mathbb{E}_{r(X^* \mid x)}[f(X^*)]$, Neighbourhood SHAP attributions add up to $f(x) - \mathbb{E}_{n(X^* \mid x)}[f(X^*)]$. Hence, care needs to be taken when comparing SHAP and Neighbourhood SHAP, since the scales might differ. In this case, both SHAP values (standard and neighbourhood) can be divided by either the sum of their absolute values or by their standard deviation, to represent relative attribution measures. As commonly observed with kernel regression approaches, there are some drawbacks, such as the additional hyperparameters (distance function, bandwidth) and increased variability especially in data sparse regions for small bandwidths. These problems can be tackled by choosing adaptive bandwidth methods. For instance, $\sigma_{nbrd}$ could be chosen such that the 25% closest observations to $x$ are not assigned more than 75% of the weight mass. We propose to plot the Neighbourhood SHAP values of the normalised features over a range of bandwidths, from $\sigma_{nbrd} = [0, 3m]$. This provides a powerful diagnostic and information tool.

The computational burden of changing $\sigma_{nbrd}$ is not as large as it might first appear. Our importance sampling approach has the desirable property that $\mathbb{E}_{n(X^*_{\bar{S}} \mid x)}[f(x_S, X^*_{\bar{S}})]$ is estimated on the same set of references $\{x^*_l\}_{l=1}^L$ for each $\sigma_{nbrd}$, and that only the importance weights vary with the bandwidth. As a result, there are no additional model evaluations required when Neighbourhood SHAP is computed for a different $\sigma_{nbrd}$. This stands in contrast to other neighbourhood schemes proposed in the XAI literature such as KDEs [9], GANs [36] or Gaussian perturbations [33] where the black box must be evaluated an additional $C \cdot L$ times for each new bandwidth where $C$ denotes the number of sampled coalitions. Please refer to Supplement C for a theoretical and empirical complexity analysis.

# 4 Smoothed SHAP

In the previous section, we discussed neighbourhood sampling as a useful tool to understand feature relevance through feature removal. We have also seen that the proposed neighbourhood sampling approach relates to kernel smoothers such as the Nadaraya-Watson estimator. This result can give us insights to consider a Smoothed SHAP that locally averages neighbouring SHAP values

$$\widehat{\phi\phi}^{smtd}(j, x) = \frac{1}{\sum_{i=1}^N d^{smtd}(x'_i, x)} \sum_{i=1}^N d^{smtd}(x'_i, x) \widehat{\phi\phi}(j, x'_i) \tag{2}$$

where $\{x_i'\}_{i=1}^N$ are samples from the reference distribution and $d^{smtd}$ is a kernel function. Such smoothing procedures have been applied before in the explainability literature, e.g. for gradient-based methods [38, 44], and can be of interest when the interpretability of SHAP values suffers under the high instability of the black box [3, 19, 21]. The smoothing it induces can be captured by a Lipschitz constant whose upper bound decreases with the bandwidth $\sigma_{smtd}$.

**Theorem 2.** *For every $x_0 \in \mathbb{R}^m$ with $||x - x_0|| < \delta$, there exists a constant $0 < L \leq max_y(f(y) - \mathbb{E}_{r(X^* \mid x)}[f(X^*)])h(\sigma_S^2)$ such that $||\phi\widehat{\phi}^{smtd}(j, x) - \phi\widehat{\phi}^{smtd}(j, x_0)|| \leq L||x - x_0||$ for the smoothed Shapley value estimator $\phi\widehat{\phi}^{smtd}(j, x)$ (2) with $d(x, x_i) = \exp(-||x - x_i||^2/\sigma_S^2)$ if $f(\cdot)$ is bounded on $\{x_i\}_{i=1}^N$ where $h(\sigma_{smtd}^2)$ is a function that decreases in $\sigma_{smtd}^2$ with $h(\sigma_{smtd}^2) \to \infty$ as $\sigma \to 0$.*

Lipschitz continuity has been established as a favourable characteristic of explainability tools [3]. Introducing smoothness can indeed lead to feature attributions that do not correctly reflect the model's behaviour. In certain circumstances, this disadvantage can be outweighed by the advantage of more "intuitive" explanations, i.e. when used to "explain to justify" [2].

With the tools from before, we can derive that Smoothed SHAP is an consistent importance sampling estimator of the SHAP values from the neighbourhood around $x$

$$\phi^{smtd}(j, x) = \underset{n(X' \mid x)}{\mathbb{E}}[\phi(j, X')] = \underset{p(S)}{\mathbb{E}}[v^{smtd}(S \cup \{j\}, x) - v^{smtd}(S, x)] \tag{3}$$

where the new value function is defined by $v^{smtd}(S, x) = \mathbb{E}_{n(X' \mid x)}[\mathbb{E}_{r(X_{\overline{S}} \mid X_S')}[f(X_S', X_{\overline{S}}^*)]]$. This smoothed value function relates to the explicit modelling of feature inclusion and gives an interesting perspective on the meaning of smoothing, namely that $x$ is a measurement of the test instance variable $X'$. Exploring a smoothed summary of the SHAP values in the local neighbourhood around $x$ highlights how local variability in $f(x)$ drives changes in the SHAP feature attributions. This is interesting in its own right but particularly so if features are susceptible to reporting error. As an illustration, consider a black box algorithm that predicts the fitness level of an adult based on multiple covariates, including weight. The reported weight may be subject to error if unreliable scales are used. In addition, as weight varies constantly throughout the day, the individual might not be interested in the attribution for one particular weight at a single point in time, but rather in the attribution that a range of weights per day receives. The test instance is thus more appropriately described by a test *distribution* of $X'$ around $x$ where $X'$ is a random variable that describes the volatility in the covariates of the test instance. If the test distribution is unknown, it can be estimated by setting it, as earlier, equal to a neighbourhood distribution $n^{smtd}(X' \mid x) \propto r^{smtd}(X' \mid x)d^{smtd}(X' \mid x)$ where $d^{smtd}(X' \mid x)$ encapsulates the prior belief on the variability of $X'$ and $r^{smtd}(X' \mid x)$ captures the artefacts of the data distribution (i.e. skew, curtosis, high density regions). The kernel $d^{smtd}(x_i', x) = \exp(-D^T(x_i', x)\Sigma_{smtd}^{-1}D(x_i', x))$ can now be defined with a multivariate bandwidth $\Sigma_{smtd} = \text{diag}(\sigma_{smtd,1}^2, ..., \sigma_{smtd,m}^2)$. We can observe empirically that such a choice can decrease the MSE of the estimation of Shapley values (Supplement K). Building upon results from kernel regression, we can quantify the squared distance of Smoothed SHAP to $\phi(j, x)$ (Supplement H). Finally, we also derive a variance estimator for Smoothed SHAP in Supplement J.

**Choice of Smoothing Bandwidth.** Prior information on the variability of the covariates of the test instance can be included in the definition of the bandwidth matrix. Fixed covariates, like age or season, are not expected to change and thus receive a bandwidth $\sigma_{smtd,j} \to 0$, while volatile features like weight, temperature or windspeed are assigned a positive bandwidth. For bandwidths $\sigma_{smtd,j} \to \infty$, the feature is treated as inherently missing. If $\sigma_{smtd,j} \to \infty$ for all features $j$, Smoothed SHAP equals the average of the Shapley values over all references which is often used as a global explanation measure [15, 11, 6]. As Smoothed SHAP can be estimated efficiently once SHAP values have been computed for the reference population, we propose, again, computing it for several bandwidth choices, and using a plot with respect to the bandwidth as a visualisation technique to help inform the choice of bandwidth. The bandwidth induces a bias-variance trade-off as derived in Supplement H: the larger the bandwidth, the smoother the results, but also the less Smoothed SHAP reflects the model behaviour at $x$, especially if $f$ is highly non-linear.

**Connection to LIME.** Tabular LIME [32, 18] provides the same explanation for any two instances falling into the same quantile along each dimension [18]. As such it is also an aggregated attribution measure, similar to Smoothed SHAP. Key differences are the treatment of different dimensions and no proven guarantees of Lipschitz continuity (see Supplement E).

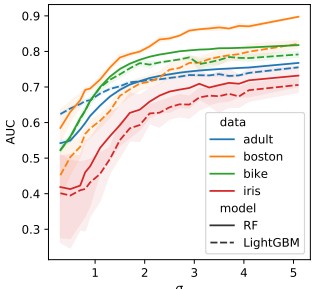
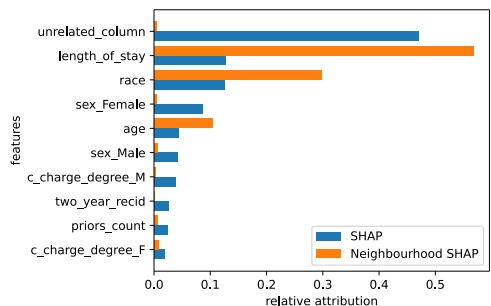

(a) AUC from OOD LightGBM and RF over 10 runs with 95% CIs. Concatenated data was created by sampling as many coalition vectors as data and masking with random references. Where references are sampled locally (smaller $\sigma_{nbrd}$), OOD classifiers perform significantly worse.

(b) Adversarial black box predicts recidivism using the COMPAS data. Absolute attributions obtained from Neighbourhood SHAP and KernelSHAP are divided by the sum of attributions for comparability. The adversarial attack affects Neighbourhood SHAP (with $\sigma_{nbrd} = 0.5$) less than KernelSHAP when averaged over 10 runs. Without adversarial attack, (Neighbourhood) SHAP attributes only `race` (not shown).

Figure 4: Neighbourhood SHAP explains on-manifold and is robust to adversarial attacks.

## 5 Examples

We present comprehensive experiments on several standardised real-world tabular UCI data sets [5] of different sizes predicted with ensemble classifiers or regressors, as well as an image classification task on the MNIST dataset. The experiments demonstrate some key attributes of Neighbourhood and Smoothed SHAP including: Neighbourhood SHAP increases on-manifold explainability and robustness against adversarial attacks; Neighbourhood SHAP also leads to sparser attributions than standard Shapley values; Smoothed SHAP tells us how Shapley values of neighbouring observations differ from the attribution of the test instance.

Since Neighbourhood SHAP, Smoothed SHAP and SHAP operate on different scales, we divide all attributions by their standard deviation (over features) unless otherwise specified. We present a subset of our results in this Section and refer the interested reader to Supplement K for a thorough report of all experimental results (including simulated experiments), details and hyper-parameter settings.

**On-Manifold Explainability and Robustness against Adversarial Attacks.** For adversarial learning, we train a Random Forest and a LightGBM as OOD classifiers that distinguish true data from concatenated vectors used for model evaluations. We find that for small bandwidths $\sigma_{nbrd}$, the adversary is not able to distinguish between the test data and the concatenated test data (Figure 4a), leading to a deterioration in their ability to discriminate true from concatenated vectors. Under the assumption that the classifiers are able to detect the true data manifold, we can thus claim that Neighbourhood SHAP relies more on observations from the data manifold than SHAP and LIME. Further, we mimicked the experimental setup of [37] on the COMPAS data set [4]: an adversarial black box predicts recidivism based only on race if the data is predicted from the OOD classifier to be from the data manifold, and returns an unrelated column if it is not. As presented in Figure 4b for 10 randomly sampled individuals, the unrelated column has no effect on Neighbourhood SHAP and race has a higher relative attribution than it does for KernelSHAP.

**Increased Local Prediction Performance.** As SHAP learns a binary feature model $g(S) = \phi_0 + \sum_{j=1}^{m} \phi_j \mathbb{I}(j \in S)$, we can sample feature coalitions and reference values to perturb test data and predict the model outcome at the perturbed data. To check local prediction performance, we weight the MSE at each test instance and at each reference value with an exponential kernel of the distance between test instance and reference value. Its bandwidth signifies the size of the neighbourhood. Figure 5 presents the MSE corresponding to an XGBoost model, applied to four different datasets. As expected, Neighbourhood SHAP with a smaller bandwidth predicts data within a small neighbourhood significantly better than Neighbourhood SHAP with a larger bandwidth. Here we noticed that the difference between the bandwidths is larger where there are fewer features in the data set (such as

the `iris` dataset). We attribute the loss in performance to the difficulty of estimating meaningful distances in high dimensions.

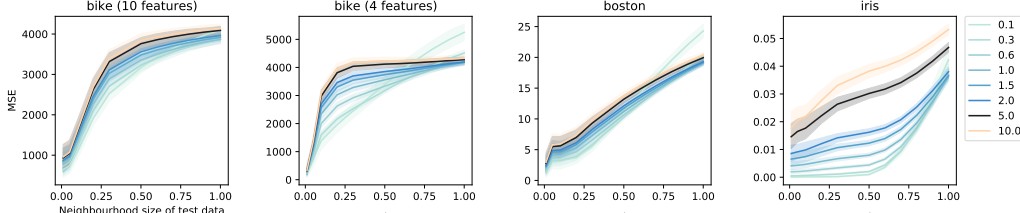

Figure 5: MSE when predicting local model outcome of an XGBoost model averaged over 400 runs displayed with 95% confidence intervals. Neighbourhood SHAP with smaller bandwidth predicts neighbourhoods significantly better than with large bandwidths.

**Interpretation of Neighbourhood SHAP.**   Neighbourhood SHAP computed with small kernel widths reflects feature attributions when contrasting with model behaviour at similar observations, whereas Neighbourhood SHAP computed with large kernel widths renders model behaviour contrasting at a population scale. Figure 6 shows the evolution of Neighbourhood SHAP across bandwidths on randomly picked observations across different data sets. The test instance in the bike data set, where a XGBoost regressor predicts daily bike rentals, has a high normalised temperature of 0.82. As the median observation has a temperature of 0.50, the neighbourhood of our test instance is expected to look considerably different to the global population. For small kernel widths, Neighbourhood SHAP computes a negative attribution for temperature, whereas marginal SHAP is positive. This sign 'flip' is coherent with descriptive statistics: for a subpopulation with temperatures +/-0.05 around 0.82, temperature is negatively correlated with outcome (correlation equal to -0.08) whereas overall, bike rental tends to increase on warmer days (unconditional correlation equal to +0.47). Neighbourhood SHAP thus shows that a warmer temperature has in general a positive impact on the count of rental bikes, which reverses for very hot days. Standard Shapley values do not provide this type of fine-grained interpretation. Similarly, in the Boston data set (Figure 6, third column), our test instance is a dwelling with a high percentage of lower status population (LSTAT) equal to 18.76%. LSTAT gets positive Neighbourhood SHAP values for small kernel widths, whereas its marginal Shapley value is negative. This observation is consistent with the negative overall correlation, which is equal to -0.76, whereas for a restricted population with LSTAT +/- 1% it is equal to +0.15. For similar dwellings i.e. with a high pupil-teacher ratio and many rooms, lower status populations do not decrease the value of the home as much as they do in general, and can even increase it.

**Interpretation of Smoothed SHAP.**   In contrast, Smoothed SHAP summarises marginal Shapley values (which contrast against the entire population) within a neighbourhood, instead of at a single instance. For example, consider the adult data set (Figure 6, first column). We chose a test instance for which the model performs poorly: its predicted probability of high income for this individual, aged 42, is equal to 0.09, when in actual fact the person has a high income. It is interesting to contrast the conventional Shapley value assigned to the person, which is obtained by Smoothed SHAP with a $\sigma_{smtd} \to 0$, with the average Shapley values for individuals like them. We observe that Smoothed SHAP quickly assigns a negative attribution to age and a positive attribution to education for $\sigma_{smtd} > 1$, whilst SHAP values were positive and negative, respectively for the individual. This highlights local instability in the Shapley values, as the SHAP numbers for people similar to the predicted person are positive for education, and negative for the age feature. For the Boston data set we note that Smoothed SHAP of the Pupil/Teacher Ratio (PTRATIO) initially decreases for a small $\sigma_{smtd}$, as there are many dwellings with a high PTRATIO in the data neighbourhood of the test instance, while it then increases as the global attribution of this feature is in general higher.

**Image Classification.**   We applied our Neighbourhood SHAP approach on KernelSHAP and also on DeepSHAP [26] which computes Shapley values for images based on gradients. After training a convolutional neural network on the MNIST data set, we explain digits with the predicted label 8 given a background data set of 100 images with labels 3 and 8. As we see in Figure 7, Neighbourhood DeepSHAP gives pixels close to the strokes attributions with the highest absolute values while DeepSHAP assigns less sparse and more blurry attributions. This is expected: DeepSHAP compares

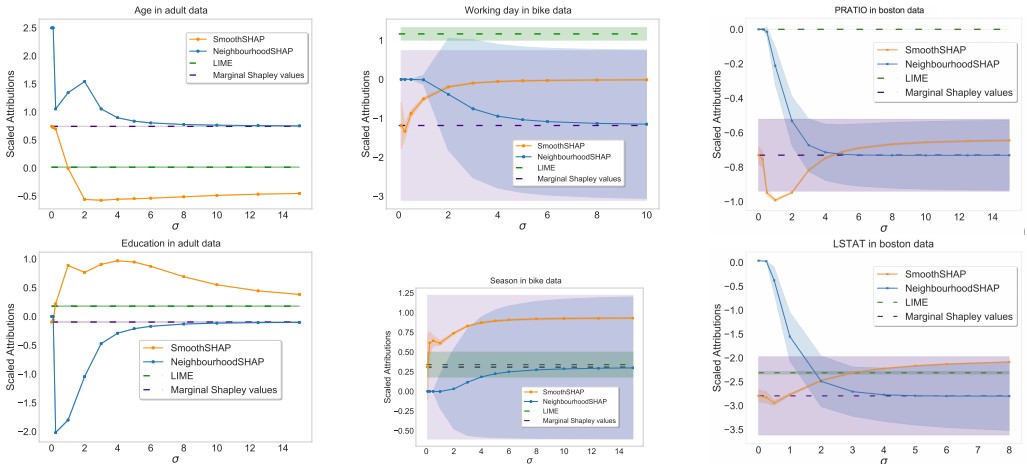

Figure 6: Scaled attributions at three different test instances (see Supplements K) for varying kernel widths computed with 2000 reference points in the adult, bike and Boston housing data sets. Bounds for LIME have been computed over 2000 runs, while the Shapley bounds have been estimated with their theoretical formula as outlined in Supplements J.

each digit to a random digit in the population, while Neighbourhood DeepSHAP only looks at images in the neighbourhood, i.e. to images which have a similar stroke. As we show in the Supplement, the change in log odds of predicting a 3 after modifying the images depicting an 8 with the attributions (setting blue pixels to 0) is (non-significantly) higher for Neighbourhood DeepSHAP than it is for DeepSHAP. In contrast, Smoothed DeepSHAP leads to a smaller change in log-odds which is expected since we lose information by smoothing. However we see that Smoothed DeepSHAP gives additional insights compared to DeepSHAP and Global DeepSHAP: In all images the lower left corner of the 8s is highlighted in blue only for Smoothed DeepSHAP. Thus, we know that there is at least one observation in the neighbourhood of these 8s that has a strong negative attribution in that image area. This image however loses importance when aggregating over the whole data set. Note that LIME gives the sharpest results because we chose the hyperparameters such that the image is split into the largest number of super pixels. We however see that LIME gives counter-intuitive results (i.e. lower right corner of the third 8 gets the lowest attribution, lower contour of first 8 gets highest attribution). Another popular explanation method called Integrated Gradients (IG) [40] was added to Figure 7 for comparison purposes. This method consists in computing the gradients along a path from the input of interest to a baseline input. Thus it resembles feature-removal based approaches in so far as it requires baseline values to be specified. The baseline is typically defined to be just the mean of the feature. If a fixed baseline value is chosen, IG suffers under the same globality problem that was outlined for SHAP.

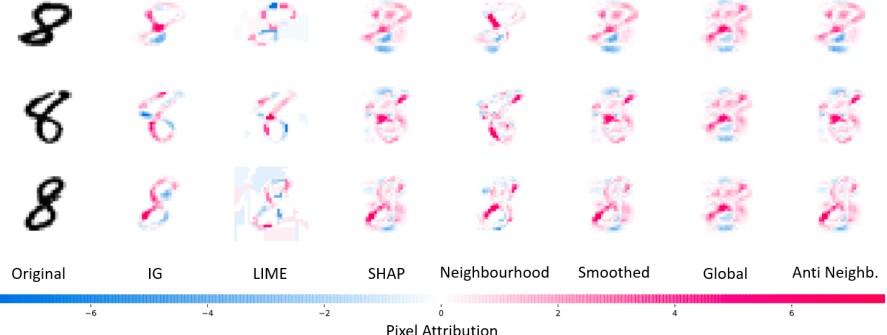

Figure 7: Randomly picked test images with explanations of the label 8. Red regions are pixels that increase the predicted probability of label 8 while blue regions decrease the predicted probability contrasted with the background data set.

**Evaluation of the proposed methods**   The two proposed methods were evaluated using a deletion metric from the ROAR framework developed by Hooker et. al [23]. For each observation, 20% of the most important features (measured by the magnitude of the absolute feature attributions) are imputed by their mean and the model is retrained. Eventually, we compare the resulting changes in predictive accuracy. We distinguish between two settings: imputation with the (global) mean of the features and imputation with the local mean (i.e. the weighted mean of each feature where the weights were computed based on a distance metric to the observation). Results are shown in the Supplementary Material (see K.7). Overall, Neighbourhood and Smoothed SHAP result in a higher absolute change in test performance than Marginal SHAP.

## 6   Discussion

In this paper, we first highlighted the limitations of using SHAP when the local model behaviour is of interest. We then introduced Neighbourhood SHAP. While neighbourhood sampling has been applied in other areas of model explainability, such as image perturbations by adding noise [14], local linear approximations (see Supplements A), or rule-based models [20, 31, 29], it has not been previously introduced for model agnostic additive feature models such as SHAP. Our contribution is important as it provides a theoretical understanding of explanations of local model behaviour, which is often lacking in the explainable AI literature [18]. A secondary contribution of this work is the analyses of how smoothing Shapley values can identify unstable feature attributions. While it is difficult to evaluate model explanations numerically, we provide an exhaustive comparison of different metrics (adversarial robustness, prediction accuracy, and visual inspection). Neighbourhood SHAP and Smoothed SHAP both merit consideration, as they have considerable advantages compared to standard KernelSHAP. For comparability across experiments, we limited our analysis to the use of the euclidean distance as a distance metric. In high dimensional spaces, this choice can be misleading [12] and the use of more powerful distance metrics, such as one obtained by random forests, would be appropriate. We thus caution against exclusively relying on mathematical metrics for explaining models, and suggest comparing the un-weighted and weighted histograms before any judgement calls. While it can be difficult to choose an adequate bandwidth, we see that having control over kernel width allows the user to have a precise understanding of model predictions, both locally and at a larger scale. LIME or KernelSHAP in their default implementation do not allow for such a detailed analysis. Plots of Neighbourhood SHAP and Smoothed SHAP w.r.t. the bandwidths $\sigma_{nbrd}$ and $\sigma_{smtd}$ respectively are thus powerful tools that give additional insight into oblique dynamics of the black box.

**Limitations and Societal Impact**   Due to the increasing use of ML-based systems to assist or to replace the human decision-maker, there is a need for transparency in algorithms. Developing robust methods that provide end-users with insightful explanations on the decisions made is key for building assurance in critical systems employing ML. However, we acknowledge that as all explainability tools, our method requires good model fit. Finally, we caution against over-reliance on mathematical metrics for explaining models, as is no consensus evaluation tool for explainability models.

## Acknowledgements

SG and LTM are students of the EPSRC CDT in Modern Statistics and Statistical Machine Learning (EP/S023151/1). SG receives funding from the Oxford Radcliffe Scholarship and Novartis. LTM receives funding from the EPSRC. KDO is funded by a Wellcome Trust/Royal Society Sir Henry Dale Fellowship 218554/Z/19/Z. CH is supported by The Alan Turing Institute, Health Data Research UK, the Medical Research Council UK, the EPSRC through the Bayes4Health programme Grant EP/R018561/1, and AI for Science and Government UK Research and Innovation (UKRI). We would like to thank Brieuc Lehmann, Luke Merrick and Daniel Moss for helpful discussions.

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
