# Supplementary Material:
# On Locality of Local Explanation Models

**Sahra Ghalebikesabi**[*]
University of Oxford
sahra.ghalebikesabi@stats.ox.ac.uk

**Lucile Ter-Minassian**[*]
University of Oxford
lucile.ter-minassian@stats.ox.ac.uk

**Karla Diaz-Ordaz**
The London School of Hygiene & Tropical Medicine & The Alan Turing Institute

**Chris Holmes**
University of Oxford & The Alan Turing Institute

## A    Local Linear Approximations vs Additive Feature Attribution Methods

In the following, we characterise two different approaches to 'local' explainability tools with neighbourhood sampling for black box models

A) *Local model approximations* where a simple model, $g(x)$, is fitted to predict the black box model, $f(x)$, in a neighbourhood around the prediction point $x$.

B) *Additive feature attribution methods* where a model-free estimator describes how the model outcome at a point $x$ changes when some of its features are removed and sampled from a reference distribution $n(x^* \mid x)$.

A black box model can be approximated with a simpler model, such as a linear function $g(x) = \sum_{j=1}^m \beta_j x_j$ at test instance $x$ by sampling reference data $\{x_i^*\}_{i=1}^L$ from a neighbourhood distribution $n(x^* \mid x)$ around $x$, and subsequently minimising the squared error loss $\mathbb{E}_{n(X^* \mid x)}[(g(X^*) - f(X^*))^2] \approx \sum_{i=1}^L (g(x_i^*) - f(x_i^*))^2$. Such an approach resembles a first-order Taylor approximation of $f$ in a local neighbourhood around $x$ defined by $n(x^* \mid x)$. It has been used for instance by [10, 2].

Thus, *local approximations* [A] capture the model's behaviour within a neighbourhood of $x$, while *additive feature attribution methods* [B] such as SHAP and LIME capture the changes in the model's outcome corresponding to $x$, comparing when a feature is included again after being removed from $x$. Although there are different flavors of removing a feature, feature removal is typically simulated by sampling a random observation $x^*$ from the statistical population given by some background data set $\{x_i\}_{i=1}^N$. The feature attribution of feature $j$ then answers the question how the model outcome changes when we move feature $j$ from a random $x_j^*$ to the test instance $x_j$. Based on the feature attributions, a linear explanation model $g(z) = \phi_0 + \sum_{j=1}^m \phi_j z_j$ can be defined where $z \in \{0, 1\}^m$.

To illustrate the difference between local linear approximations and additive explanation models, consider explaining a black box $f(x) = \mathbb{I}(x_1 > 0)2x_2^2 - \mathbb{I}(x_1 \le 0)x_2^2$ given i.i.d. realisations $\{x_i\}_{i=1}^N$ of $X \sim \text{Normal}(0, 1)$. Results are depicted in Figure 1. As we see the attribution of Feature-1 computed by SHAP and LIME is constant no matter how far we are from the decision boundary. This result is expected since for fixed Feature-2 the change in model outcome is constant for removing Feature-1 whenever Feature-1 is positive (or negative respectively). [9] define *locality* as the aim to understand a prediction by asking 'if the input is changed slightly, how does the model's

---

[*]equal contribution

35th Conference on Neural Information Processing Systems (NeurIPS 2021).

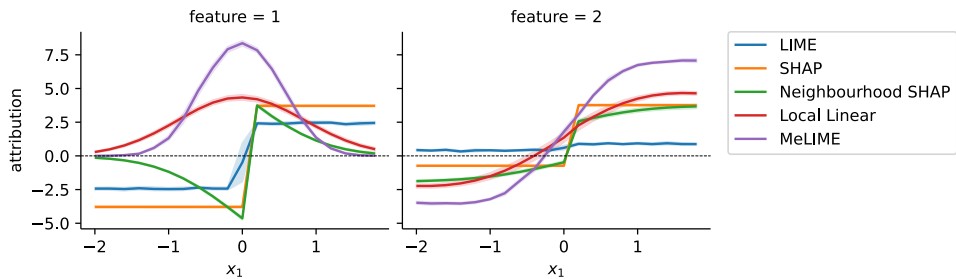

Figure 1: Attributions at $x = (x_1, 2)$ with $x_1$ varying for a reference distribution of $X \sim$ Normal$(0, 1)$ and black box $f(x) = \mathbb{I}(x_1 > 0)2x_2^2 - \mathbb{I}(x_1 \leq 0)x_2^2$ averaged over 10 runs displayed with 95% confidence intervals (see next section for details). While (Tabular) LIME and SHAP assign the same absolute attribution to Feature-1 no matter how large $x_1$ is, our neighbourhood approach takes its distance to the decision boundary into consideration. A local linear approximation to the black box trained with an euclidean distance based exponential kernel weighted Ridge Regressor, i.e. on the same neighbourhood as Neighbourhood SHAP, gives misleading attributions to Feature-1 for $-0.4 < x_1 < 0$.

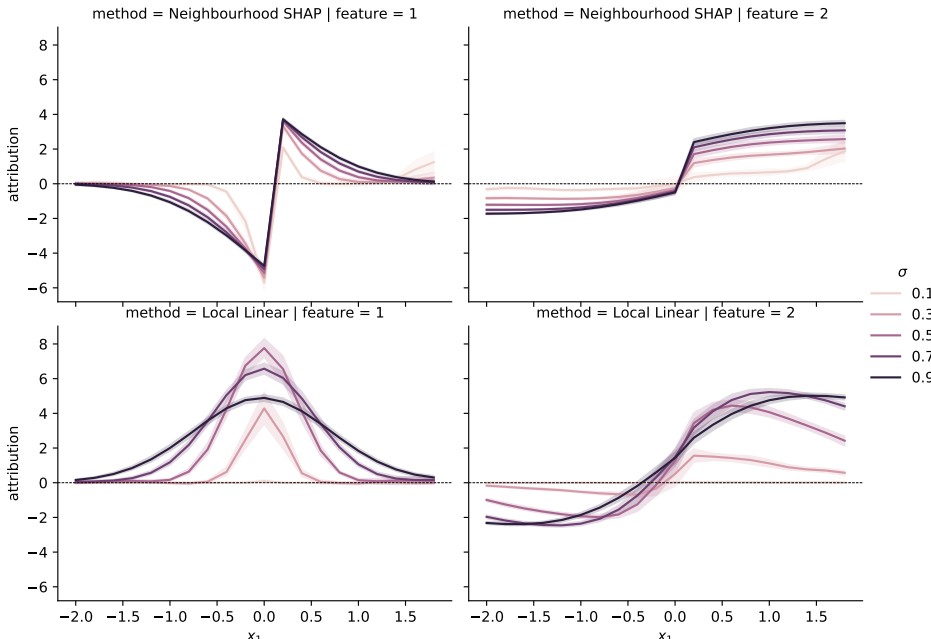

Figure 2: Feature attributions for the earlier example for different bandwidths (differentiated by the hue) for Neighbourhood SHAP (first row) and a local linear approximation (second row). As we see, the local linear model has a higher variance in general. We also note that the local linear attribution of Feature-2 goes to 0 for larger values of $x_1$ which is not desired. This results from the fact that neighbourhoods around $x$ are sparser the larger $x$ gets. For $\sigma = 0.1$, we note that Neighbourhood SHAP also suffers under high variance in the Feature-2 attribution for extreme values.

prediction change?'. As this example shows, LIME and SHAP capture global patterns of the black box model, despite being referred to as 'local' explanation models. This observation motivates the use of Neighbourhood SHAP. A linear approximation model such as MeLIME fits a linear approximation around $x$. Since at small negative values for Feature-1, observations with a positive value for Feature-1 are also included in the neighbourhood, the linear approximation is influenced by the higher positive effect of the black box and thus attributes the feature positively.

Note that LIME is a general framework that has been defined as additive feature attribution method (also see Supplement E). However, its implementation allows to also train a linear approximation

by sampling data from the reference distribution, weighting it with its distance to the test instance $x$, and training a linear model such as Ridge on the weighted data. Some extensions such as MeLIME [2] or MAPLE [10] extend LIME by proposing more meaningful neighbourhood sampling schemes. However they only consider a linear approximation setting. It has been noted that LIME underperforms compared to linear approximation methods [9, 2] when performance is measured by prediction accuracy within a small neighbourhood around $x$. This is not surprising as the Tabular version of LIME as additive feature attribution method (which the papers compare to) is not defined by taking local neighbourhoods into consideration.

In contrast to local linear approximations, our proposal, Neighbourhood SHAP, increases the 'locality' of SHAP by sampling the removed features from a neighbourhood around $x$, instead of from the (global) statistical population. There are several reasons why a user might be more interested in an additive feature attribution (referred to as method [B]) that builds upon neighbourhood distributions than a local linear approximation (referred to as method [A]):

- Approach [A] requires the subjective choice of the parametric local model $g(x)$. As has been pointed out elsewhere [12], if $g(x)$ is a good approximation to $f(x)$ for all $x \sim X$ then the user should adopt $g(x)$ as their preferred model (as it is explainable). If $g(x)$ is not a faithful local representation of $f(x)$ then it is challenging to trust its interpretability.

- Approach [A] translates the local behaviour of $f(x)$ through $g(x)$ and hence any interpretation and statement of explanations must be contextualised in light of this, as the translator's version of $f(x)$. Instead approach [B], and in particular Neighbourhood SHAP, has an interpretation in terms of the expected change in $f(x)$ that does not require a surrogate model.

- Approach [B] is model free, i.e. it does not make any parametric assumptions. The method attributes changes in the local expectations of $f(x)$ to features of $x$ through a sum-of-squares (variance) decomposition [8].

Taken together we believe approach [B] has strong merit as a local explainability measure.

## B Axioms of Shapley Values

Shapley values have been shown to satisfy the following axioms.

- According to the *Dummy* axiom, a feature $j$ receives a zero attribution if it has no possible contribution, i.e. $v(S \cup j) = v(S)$ for all $S \subseteq \{1, ..., m\}$.

- According to the *Symmetry* axiom, two features that always have the same contribution receive equal attribution, i.e. $v(S \cup i) = v(S \cup j)$ for all $S$ not containing $i$ or $j$ then $\phi_i(v) = \phi_j(v)$.

- According to the *Efficiency* axiom, the attributions of all features sum to the total value of all features. Formally, $\sum_j \phi_j(v) = v(\{1, .., m\})$.

- According to the *Linearity* axiom, for any value function $v$ that is a linear combination of two other value functions $u$ and $w$ (i.e. $v(S) = \alpha u(S) + \beta w(S)$), the Shapley values of $v$ are equal to the corresponding linear combination of the Shapley values of $u$ and $w$ (i.e. $\phi_i(v) = \alpha \phi_i(u) + \beta \phi_i(w)$).

Since all these axioms have been defined conditionally on the value function $v$, they hold if the value function is changed. We have expressed both Neighbourhood SHAP and Smoothed SHAP as a change in the value function of standard Shapley values. As such they also adhere to the axioms.

## C Computational Burden

In this section, we want to illustrate the computational complexity of Neighbourhood SHAP. Please refer to Figures 3 for plots that describe the additional computational complexity of computing the proposed SHAP values.

**Computation using the Shapley formula.** One way of computing Shapley values is to use a mean estimator. For marginal reference distributions, it holds in particular that

$$\phi(j, x) = \mathop{\mathbb{E}}_{r(X_{\overline{S}}^* \mid x)} \left[ \mathop{\mathbb{E}}_{S}[f(x_{S \cup j}, X_{\overline{S} \setminus j}^*) - f(x_S, X_{\overline{S}}^*)] \right] = \mathop{\mathbb{E}}_{x^* \sim r(X_{\overline{S}}^* \mid x)} [\phi_{x^*}(j)].$$

Then,

$$\phi_{x^*}(j) = \mathop{\mathbb{E}}_{S}[f(x_{S \cup j}, x_{\overline{S} \setminus j}^*) - f(x_S, x_{\overline{S}}^*)] \tag{1}$$

is referred to as single-reference Shapley value [7], which can be characterised as the expected change in model outcome when feature $j$ of observation $x^*$ is set equal to $x_j$ after a random subset of features $x_S^*$ has already been set equal to $x_S$. When the reference distribution is marginal, we can write the local Shapley value as a weighted average of the single-reference Shapley values

$$\hat{\phi}^{local}(j) = \frac{1}{\sum_{i=1}^{L} d(x_{i,\overline{S}}^* \mid x_{\overline{S}})} \sum_{i=1}^{L} d(x_{i,\overline{S}}^* \mid x_{\overline{S}}) \phi_{x_i^*}(j)$$

where $\{x_i^*\}_{i=1}^L$ are samples from $r(x^* \mid x)$. Computing Neighbourhood SHAP is then just a linear transformation of the single-reference Shapley values. As such the complexity for computing Neighbourhood SHAP for any additional $\sigma_N$ for all features is $\mathcal{O}(L \cdot m)$: after the computation of $L$ weights, these have to multiplied with the single-reference Shapley values and the weighted values are added up.

**Computation using KernelSHAP.** The KernelSHAP optimisation of

$$\mathop{\mathbb{E}}_{S}[(\mathop{\mathbb{E}}_{r(X_{\overline{S}}^* \mid x)}[f(x_S, X_{\overline{S}}^*)] - g(S))^2] \approx \sum_{l=1}^{L} \sum_{i=1}^{C} w_i (f(x_{S_i}, x_{l,\overline{S}_i}^*) - g(S_i))^2 \tag{2}$$

can be simplified to

$$\mathop{\mathbb{E}}_{S}[(\mathop{\mathbb{E}}_{r(X_{\overline{S}}^* \mid x)}[f(x_S, X_{\overline{S}}^*)] - g(S))^2] \approx \sum_{i=1}^{C} w_i (\frac{1}{L} \sum_{l=1}^{L} f(x_{S_i}, x_{l,\overline{S}_i}^*) - g(S_i))^2. \tag{3}$$

As such we only add weighting to the optimisation

$$\mathop{\mathbb{E}}_{S}[(\mathop{\mathbb{E}}_{r(X_{\overline{S}}^* \mid x)}[f(x_S, X_{\overline{S}}^*)] - g(S))^2] \approx \sum_{i=1}^{C} w_i (\sum_{l=1}^{L} \frac{d(x_l^* \mid x)}{\sum_{k=1}^{L} d(x_k^* \mid x)} f(x_{S_i}, x_{l,\overline{S}_i}^*) - g(S_i))^2. \tag{4}$$

Computing the weights can be done in $\mathcal{O}(L \cdot m)$, while the cost for aggregation of the model outcome over coalitions is an additional $\mathcal{O}(L \cdot C)$. Finally, the optimisation using matrix multiplication is of complexity $\mathcal{O}(C \cdot m)$. All in all, the computational cost is thus $\mathcal{O}(\max(L, m) \cdot C)$.

**Time Complexity of Smoothed SHAP.** Once $N$ Shapley values have been computed, Smoothed SHAP can be computed in $\mathcal{O}(N \cdot m)$.

# D   Anti-Neighbourhood SHAP

Among others, Shapley values satisfy the linearity axiom [6] which says that two attributions $\phi_v(j), \phi_w(j)$ with value functions $v, w$ add up to $\phi_{v+w}(j)$. As a result, we can characterise the difference between the marginal Shapley values $\phi_{\mathbb{E}_{r(X^* \mid x)}[f(x_S, X_{\overline{S}}^*)]}$ and the Neighbourhood Shapley values $\phi_{\mathbb{E}_{n(X^* \mid x)}[f(x_S, X_{\overline{S}}^*)]}$ as Anti-Neighbourhood Shapley values $\phi^{anti} = \phi_{\mathbb{E}_{r(X^* \mid x)}[f(x_S, X_{\overline{S}}^*)] - \mathbb{E}_{n(X^* \mid x)}[f(x_S, X_{\overline{S}}^*)]}$

$$\mathop{\mathbb{E}}_{r(x^* \mid x)} [f(x_S, x_{\overline{S}}^*)] - \mathop{\mathbb{E}}_{n(x^* \mid x)} [f(x_S, x_{\overline{S}}^*)] \approx \frac{1}{L} \sum_{i=1}^{L} f(x_S, x_{i,\overline{S}}^*) \left( 1 - \frac{d(x_{i,\overline{S}}^* \mid x_{\overline{S}})}{\frac{1}{L} \sum_{l=1}^{L} d(x_{l,\overline{S}}^* \mid x_{\overline{S}})} \right).$$

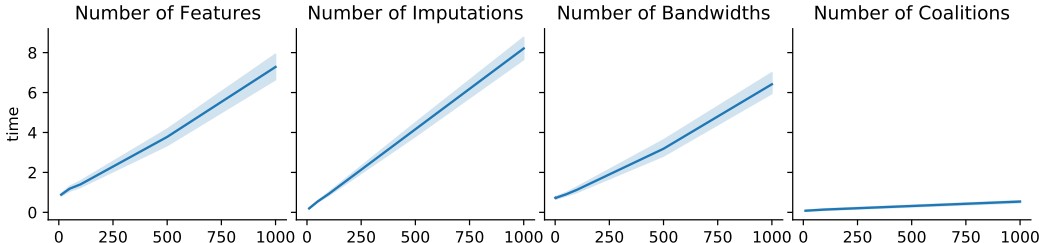

Figure 3: Plots of the computational time (seconds in clock time) to compute Neighbourhood SHAP w.r.t the number of imputations $L$ (default: 100), the number of features $m$ (default: 11), the number of coalitions $C$ (default: $2^{11}$), and the number of bandwidths (default: 50) averaged over 10 runs displayed with 95% confidence intervals, run on a 2.4 GHz 6-Core Intel Core i5-9300H CPU, using the SHAP package [6].

While Neighbourhood SHAP computes the expected change in model outcome when the features of a random observation in the neighbourhood of $x$ are set equal to $x$, Anti-neighbourhood SHAP weights the change in model outcome higher for observations that are farther away from $x$. This might be of interest, if the user is worried about loss in information from only using Neighbourhood SHAP. Consider the black box from Supplement A. Even though Feature-1 has no local effect on the black box for large values of Feature-1, it affects the value of the test instance globally: this is reflected by Anti-Neighbourhood SHAP.

## E   Explaining LIME

LIME [11] is defined as an optimisation problem in the binary coalition space

$$\epsilon(x) = \operatorname{argmin} \sum_{z,z' \in \mathcal{Z}} d(z|x)(f(z) - g(z'))^2 + \Omega(g) \tag{5}$$

where $\Omega(g)$ is a penalty on the complexity of $g$, i.e. an $l1$ loss, $z$ are samples drawn from the feature space, and $z'$ are the corresponding binary representations. While the computation of LIME differs depending on the data type (i.e. image, text or tabular data), we will focus on tabular data here, as its corresponding implementation is used for comparisons on simulated data sets [2] or to determine predictive accuracy [10].

In order to compute the attribution at a local observation $x$, Tabular LIME follows the following steps:

1. Given a background data set, learn the $k$ quantiles $q_1, ..., q_k$ of the data distribution with their summary statistics (mean and standard deviation).
2. Compute each dimension $j$ of each imputation $i$, $z_{i,j}$, with $i \in \{1, ..., L\}, j \in \{1, ..., m\}$ as follows
   a) Sample a quantile $b_{i,j}$ for each feature $j$ by uniformly sampling from $\{q_1, ..., q_k\}$.
   b) Sample an observation $z_{i,j}$ from a truncated Gaussian fitted with the summary statistics of quantile $b_{i,j}$ for each feature $j$.
   c) Turn the observation $z_i$ into its discretised mapping $z_i'$ by setting $z_{i,j}$ equal to 1 if feature $j$ of the test instance, i.e. $x_j$, falls in quantile $b_{i,j}$, and 0 otherwise.
3. Solve the optimisation problem in Equation 5 with $d(z|x)$ replaced by $d(z'|x')$ where $x' = 1_m$ is a vector of all ones as $x_j$ always falls in the same bin as $x_j$.

For a more in depth depiction of the algorithm, we refer the reader to [4]. Note that any observation $u$ where each dimension falls into the same bin as the corresponding dimension from $x$, i.e. $q(u_j) = q(x_j)$ where $q$ returns the quantile of its argument, gets the same attribution as $x$. As such, LIME is an aggregated attribution measure. However only observations that fall into the same bin in each dimension get the same attribution. As such it uses naive histogram weights, i.e. $I(q(z_j) = q(x_j)$ for all $j$) To analyse the smoothness of the LIME attributions, let the lower bound of $q$ be

| Feature | 1 | 2 | 3 |
|---|---|---|---|
| LIME | 0.028 | 0.018 | 0.624 |
| SHAP | 0.090 | 0.090 | 0.819 |
| (Ours) Neighbourhood SHAP ($\sigma = 0.5$) | 0.063 | 0.074 | 0.079 |

Table 1: Attributions at $x = (1.001, 1.001, 1.001)$ for a reference distribution of $X \sim$ Uniform$(-2, 2)$ and black box $f(x) = \mathbb{I}(x_1 > 1 \text{ or } x_2 > 1)x_3$ computed over 10 runs.

denoted by $q_l$ while the upper bound is denoted by $q_u$. Now consider two observations $x$ and $u$ with $x_j = q_{1,l} + \frac{1}{2}\sqrt{\epsilon/m}$ and $u_j = q_l(x_j) - \frac{1}{2}\sqrt{\epsilon/m}$ for each feature $j$ for an arbitrary $\epsilon$. It then follows that $\|x - u\| \leq \epsilon$. Since we cannot make any statement about the difference in attributions of any two such observations without additional assumptions, Lipschitz continuity does not have to hold necessarily. Note that a difference to Smoothed SHAP is that the smoothing is applied to each dimension independently.

Some other interesting properties of this algorithm include:

- The reference values $z$ are sampled from a global reference distribution which consists of a mixture of non-overlapping truncated Gaussian distributions where each mixture component has the same probability.

- Each dimension of $z$ is sampled independently.

- The probability of a coalition $z'$, or as earlier defined by $S$, is $P(S) \propto \exp(-(|S|-m)^2/\sigma^2)/k^{|S|}$ where $k$ is the number of quantiles.

Note that often LIME is understood as sampling local data by weighting it with a distribution that depends on the distance in the data space [14, 2]. However, interestingly enough this is not how Tabular LIME is implemented by [11].

Instead, observations with a close coalition representation $z'$ are weighted higher. This does not ensure that close observations receive a higher importance. Consider a three dimensional feature space and a test instance of $x = (1, 1, 1)$, the model outcome $f(1, 1, 10000)$ for coalition $S = \{1, 2\}$ will be weighted higher in the optimisation than the model outcome $f(1, 2, 2)$ for coalition $S = \{1\}$. This enforced locality in the coalition space of LIME will lead to lower attributions of a feature if its effect is reduced by the presence of another feature. Consider for instance a simple rule based model such as $f(x) = \mathbb{I}(x_1 > 1 \text{ or } x_2 > 1)x_3$ at $x = (1.001, 1.001, 1.001)$. Perturbing, *either* Feature-1 *or* Feature-2 does not have any effect on the model outcome. For a reference distribution of $X \sim$ Uniform$(-2, 2)$, the attributions of $x_1$ and $x_2$ are therefore almost zero (Table 1). In contrast, Shapley values find a relatively higher attribution for Features-1 and 2, but still a considerably higher attribution for Feature-3. In a small neighbourhood around $x$, this attribution can be misleading.

## F   Nadaraya-Watson Estimator

We show that the Nadaraya-Watson estimator can be interpreted as an importance sampling estimator. In kernel regression the aim is to model the non-linear relationship between a dependent variable $Y$ and an independent variable $Z$, by approximating the conditional expectation $\mathbb{E}[Y \mid Z]$. This is traditionally argued for by an approximation of the conditional density $p(y|z)$ by estimating both $p(z, y)$ and $p(z)$ with kernel regressors, i.e.

$$\mathbb{E}[Y|Z = z] = \int y \cdot p(y|z)dy = \int y \frac{p(z, y)}{p(z)} dy \text{ with estimators } \hat{p}(x) = \frac{1}{L}\sum_{i=1}^{L} d(z_i|z),$$

$$\hat{p}(z, y) = \frac{1}{L}\sum_{i=1}^{L} d(z_i|z)d(y_i|y) \text{ and thus } \hat{\mathbb{E}}[Y|Z = z] = \frac{\sum_{i=1}^{L} d(z_i|z)y_i}{\sum_{j=1}^{L} d(z_j|z)}.$$

# G   Lipschitz Continuity of Smoothed SHAP

For a data set $\{x_i\}_{i=1}^N$ and bounded black box $f$, we show that Smoothed SHAP increases Lipschitz continuity. To ensure this, for any two values $z, y \in \mathbb{R}^m$ with $k \in \{1, \ldots, m\}$ it has to hold that

$$\left\| \hat{\phi}^{smoothed}(k, z) - \hat{\phi}^{smoothed}(k, y) \right\|^2 \leq L^2 \|z - y\|^2$$

for a constant $L$. Note that not any bounded function is Lipschitz continuous, and that this is thus a non-trivial proof.

For simplicity we drop the index $k$ in the following. Let $\delta^2 = \|z - y\|^2$. Because of the triangle inequality it follows that $\|x_i - z\|^2 \leq \|x_i - y\|^2 + \delta^2$. Building upon this, we derive that

$$\left\| \hat{\phi}^{smoothed}(z) - \hat{\phi}^{smoothed}(y) \right\|$$

$$= \left\| \frac{1}{\sum_{j=1}^N d(x_j \mid z)} \sum_{i=1}^N \hat{\phi}(x_i) d(x_i \mid z) - \frac{1}{\sum_{j=1}^N d(x_j \mid y)} \sum_{i=1}^N \hat{\phi}(x_i) d(x_i \mid y) \right\|$$

$$= \left\| \sum_{i=1}^N \hat{\phi}(x_i) \left( \frac{d(x_i \mid z)}{\sum_{j=1}^N d(x_j \mid z)} - \frac{d(x_i \mid y) \sum_{j=1}^N d(x_j \mid z)}{\sum_{j=1}^N d(x_j \mid y) \sum_{j=1}^N d(x_j \mid z)} \right) \right\|$$

$$= \left\| \sum_{i=1}^N \hat{\phi}(x_i) \left( \frac{\exp(-\|x_i - z\|^2 / \sigma^2)}{\sum_{j=1}^N \exp(-\|x_j - z\|^2 / \sigma^2)} \right. \right.$$
$$\left. \left. - \frac{\exp(-\|x_i - y\|^2 / \sigma^2) \sum_{j=1}^N \exp(-\|x_j - z\|^2 / \sigma^2)}{\sum_{j=1}^N \exp(-\|x_j - y\|^2 / \sigma^2) \sum_{j=1}^N \exp(-\|x_j - z\|^2 / \sigma^2)} \right) \right\|$$

$$\leq \left\| \sum_{i=1}^N \hat{\phi}(x_i) \left( \frac{\exp(-\|x_i - z\|^2 / \sigma^2)}{\sum_{j=1}^N \exp(-\|x_j - z\|^2 / \sigma^2)} \right. \right.$$
$$\left. \left. - \frac{\exp(-\|x_i - z\|^2 / \sigma^2) \exp(-\delta^2/\sigma^2) \sum_{j=1}^N \exp(-\|x_j - y\|^2 / \sigma^2) \exp(-\delta^2/\sigma^2)}{\sum_{j=1}^N \exp(-\|x_j - y\|^2 / \sigma^2) \sum_{j=1}^N \exp(-\|x_j - z\|^2 / \sigma^2)} \right) \right\|$$

$$= \left\| \sum_{i=1}^N \hat{\phi}(x_i) \frac{\exp(-\|x_i - z\|^2 / \sigma^2)(1 - \exp(-2\delta^2/\sigma^2))}{\sum_{j=1}^N \exp(-\|x_j - z\|^2 / \sigma^2)} \right\|$$

$$= \left\| \hat{\phi}^{smoothed}(z) \overbrace{1 - \exp(-2\delta^2)}^{\leq \delta} \right\|$$

$$\leq \left\| \underbrace{\hat{\phi}^{smoothed}(z) \frac{1}{\sum_{j=1}^N \exp(-\|x_j - z\| / \sigma^2)}}_{L = max_x \left( \hat{\phi}^{smoothed}(x) \right)} \delta \right\|$$

$L$ exists as long as any Shapley value is bounded which follows from the bound on $f(\cdot)$. The inequality $1 - \exp(-2\delta^2) \leq \delta$ holds trivially for $\delta \geq 1$. For $0 < \delta < 1$, it holds that

$$1 - \exp(-2\delta^2) < \delta$$
$$-2\delta^2 > log(1 - \delta)$$
$$-2\delta^2 > -\sum_{k=1}^{\infty} \frac{\delta^k}{k}$$
$$\delta^2 < \frac{1}{2} \sum_{k=1}^{\infty} \frac{\delta^k}{k}$$
$$1 < \frac{1}{2\delta} + \frac{1}{4} + \frac{\delta}{6} < \frac{1}{2} \sum_{k=1}^{\infty} \frac{\delta^{k-2}}{k}$$

The last inequality follows from the fact that the roots of the polynomial $\frac{1}{2} + \frac{1}{4}\delta + \frac{\delta^2}{6}$ are outside of $(0, 1)$.

## H   Smoothed SHAP as Kernel Regressor

When smoothing is formulated as a kernel regressor, we can use this relationship to quantify the bias and the variance of the smoothed estimator from the true Shapley values building upon results for the multivariate Nadaraya-Watson estimator, as we will do in the following.

Assuming that $\hat{\phi}(j, x_i)$ for $i \in \{1, \ldots, N\}$ was computed as a mean over single-reference Shapley values 1, that for each $i$ the background data set has been sampled i.i.d., and that the black box $f$ is bounded, then $\hat{\phi}(j, x_i)$ is a mean estimator and it follows from the Central Limit Theorem for a large number of references $L \to \infty$ that

$$\hat{\phi}(j, x_i) \approx \phi(j, x_i) + \epsilon_i \tag{6}$$

where $\epsilon_i \sim \text{Normal}(0, \sigma_{ref}^2/L)$ denotes i.i.d. normal noise with $\sigma_{ref}^2$ being the variance of the single-reference Shapley values. Note that this form (with possibly different noise distribution) follows for each unbiased estimator $\hat{\phi}$ of $\phi$. Since KernelSHAP is assumed to be biased, a bias-corrected version such as the one proposed by [3] can be used.

Let $\Sigma$ denote the matrix of bandwidths, i.e. here $\Sigma = \text{diag}(\sigma_1, \ldots, \sigma_m)$. We then make the following assumptions

1. The Shapley estimator is unbiased, i.e. Equation 6 holds.

2. The variance of $\text{Var}[\hat{\phi}(j, x) \mid x] =: \sigma_{\hat{\phi}}^2$, is continuous and positive.

3. The test instance $x$ lies in the support of $f$. The probability density function of $\hat{\phi}(j, x)$, $p(\hat{\phi}(j, x))$, is continuously differentiable and bounded away from zero.

4. The kernel, i.e. density $d(x^*|x)$, is a symmetric and bounded PDF with finite second moment and square integrable, such as the exponential kernel.

5. The sequence of $\Sigma$ is a deterministic sequence of positive definite symmetric matrices such that $N^{-1}|\Sigma|$ and each entry of $\Sigma$ tend to 0 when $n \to \infty$.

Note that the first two assumptions hold for Shapley values computed with KernelSHAP or the Shapley formula, as long as $f$ has a finite expectation. It then holds that [13]

$$Bias[\hat{\phi}^{smoothed}(j, x), \phi(j, x)] = \frac{1}{2}\mu_2(d) \operatorname{tr}\left(\Sigma \mathcal{H}_{\phi(j)}(x)\right) + o\left(\operatorname{tr}(\Sigma)\right)$$

$$\text{Var}[\hat{\phi}^{smoothed}(j, x) \mid x_1, \ldots, x_N] = \frac{1}{N\sigma_1 \cdot \ldots \sigma_m} \|d\|_2^2 \frac{\sigma_{\hat{\phi}}^2}{p(\hat{\phi}(j, x))} \left(1 + o(1)\right))$$

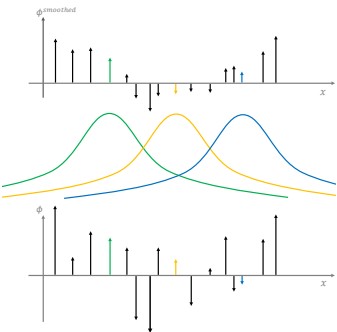

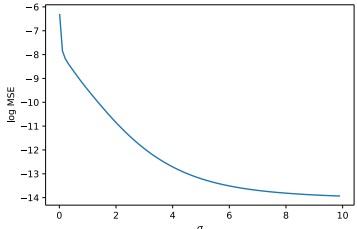

(a) The coloured exponential kernels illustrate how the smoothed Shapley values can be computed in a moving average like manner from the standard Shapley values.

(b) Log MSE for the black box $f(x) = x$ with $X \sim$ Normal$(0, 1)$ with 100 reference points for 1000 instances. The smoothed Shapley values were corrected by their offset $f(x) - \mathbb{E}_{p(X)}[f(X)]$.

Figure 4: Illustrations of Smoothed SHAP.

where $\mu_2 = \int uu^T d(u|0)du$, $\|d\|_2^2 = \int d(s|0)^2 ds$, $\mathcal{H}_f$ is the Hessian matrix of $f$, and $\mathrm{tr}(H)$ is the trace of the matrix $H$. For an exponential kernel it is $\|d\|_2^2 = 2^{-m}\pi^{-m/2}$. The Hessian $\mathcal{H}_{\phi(j)}$, as a measure of the complexity of the Shapley value function $\phi(j)(x) := \phi(j, x)$, in the bias term signalises that the bias increases the more instable Shapley values are to small perturbations of $x$. For smooth functions, i.e. linear functions, smoothing can indeed decrease the bias as seen in Figure 4b.

## I   Modelling Measurement Error with Smoothed SHAP

As explained in the main part of the paper, Smoothed SHAP relates to modelling feature inclusion which allows to take the volatility in the measurement of the test instance into consideration.

In a first simulated example, we assume a one-dimensional test instance $x$ is measured with standard normal measurement error and linear black box, as presented in Figure 5. Since the measurement error is attributed to the only present feature, observations with close true value $x'$ can have highly varying attributions because of the measurement error in $x$.

We will now illustrate Smoothed SHAP in the context of real-world examples. For simplicity we consider the prediction task with a boosted regressor on the bike data set with four features: `season`, `month`, `hour` and `humidity`. In a first scenario, we assume that we do not have access to the hourly humidity of the test instances, but only to their daily average. In a second case, we assume that the humidity measurement of the test instances is noisy, i.e. perturbed with normal noise of scale 0.1. Smoothed SHAP was then computed as a weighted average over the Shapley values of the training data and the SHAP values of the test data with measurement error. We set the bandwidths of the first three features close to 0 such that Smoothed SHAP of $x'$ only depends on observations $x$ that equal $x'$ in the first three dimensions. The bandwidth of `hour` was tuned on a validation set. We also repeated a similar set up on the adult data set: a boosted classifier predicted high income based on the features `Workclass`, `Education-Num`, `Marital Status` and `Age`. In the test data, we do not have access to the age of an individual, but only to his age group (0-10, 10-30, 30-50, 50+). Again we only average over the SHAP values of individuals with the same first features as the test instance. Results are presented in Table 2.

## J   Variance Analysis

Shapley values can be estimated by

$$\hat{\phi}(j, x) = \frac{1}{m} \sum_{S \subseteq \{1,\ldots,m\}/j} \left[ \frac{1}{\binom{m-1}{|S|}} \left( \frac{1}{L} \sum_{i=1}^{L} f(x_{S \cup j}, x^*_{i, \overline{S}/j}) - \frac{1}{L} \sum_{i=1}^{L} f(x_S, x^*_{i, \overline{S}}) \right) \right].$$

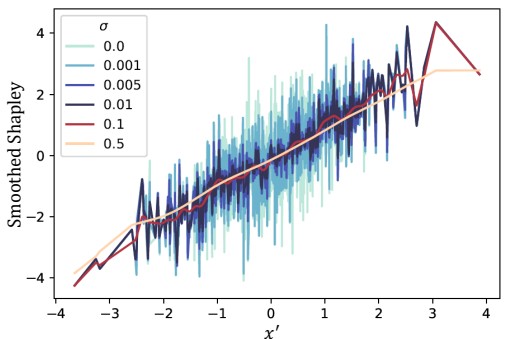

| | MSE |
|---|---|
| bike - daily average as measurement | |
| SHAP | 291.8637±0.8304 |
| Smoothed SHAP | **287.7668±0.8195** |
| bike - noisy measurement | |
| SHAP | 195.0134±0.6174 |
| Smoothed SHAP | **190.6908±0.6026** |
| adult - age groups | |
| SHAP | 0.0037±0.0000 |
| Smoothed SHAP | **0.0030±0.0000** |

Figure 5: Smoothed SHAP for different bandwidths (where $\sigma = 0$ denotes the standard marginal Shapley values) for $f(x) = x = x' + \epsilon$ and $X', \epsilon \sim \mathrm{Normal}(0,1)$ for 1000 imputations and exponential kernel. Without smoothing, SHAP is misleading for the test instance $x$ as the noise in measurement is attributed to the only present feature.

Table 2: MSE±standard error averaged over 5,000 runs on bike data. The bandwidth for the humidity feature of Smoothed SHAP was tuned on a validation set (20% of train data as described in Supplement K) over a range of 0.5 and 2. We see that Smoothed SHAP has always a significantly smaller MSE.

We note that $\phi(j, x)$ is the expectation of a discrete variable $V \mid \{X^*_{i,\overline{S}/j}, X^*_{\overline{S}}\}_{S \subseteq \{1,\ldots,m\}/j}$ with

$$V \mid \{X^*_{\overline{S}/j}, X^*_{\overline{S}}\}_{S \subseteq \{1,\ldots,m\}/j} = \mathop{\mathbb{E}}_{r(X^*_{\overline{S}/j} \mid x)} \left[ f(x_{S \cup j}, X^*_{\overline{S}/j}) \right] - \mathop{\mathbb{E}}_{r(X^*_{\overline{S}} \mid x)} \left[ f(x_S, X^*_{\overline{S}}) \right]$$

with a support of size $2^{m-1}$ and probabilities $\left\{ \frac{1}{m\binom{m-1}{|S|}} \text{ for } S \subseteq \{1,\ldots,m\}/j \right\}$. Since the support of this discrete variable is estimated using a mean estimator, we can compute the variance of the estimated Shapley value as

$$\mathrm{Var}[\hat{\phi}(j)] = \frac{1}{m^2} \sum_{S \subseteq \{1,\ldots,m\}} \left[ \frac{1}{\binom{m-1}{|S|}^2} \left( \frac{\sigma^2_{S \cup j}}{L} + \frac{\sigma^2_S}{L} \right) \right]$$

by independence and $\mathrm{Var}[aX] = a^2 \mathrm{Var}[X]$ where $\sigma^2_S$ is the variance of $f(x_S, X^*_{\overline{S}})$ which we estimate with the squared standard deviation of $\{f(x_S, x^*_{i,\overline{S}})\}_{i=1}^L$. For computing the variance of Neighbourhood SHAP, we use results from self-normalised importance sampling (see [5] or http://statweb.stanford.edu/~owen/mc/Ch-var-is.pdf) and estimate the variance $\sigma^2_S$ now by

$$\hat{\sigma}^2_S = \frac{1}{(\sum_{k=1}^L d(x_S | x^*_{k,S}))^2} \sum_{i=1}^L d(x_S | x^*_{i,S})^2 (f(x_S, x^*_{i,\overline{S}}) - \bar{f}_w(x_S, x^*_{i,\overline{S}}))^2$$

where $\bar{f}_w(x_S, x^*_{i,\overline{S}})$ denotes the weighted average of $\{f(x_S, x^*_{i,\overline{S}})\}_{i=1}^L$ with weights $\left\{ \frac{d(x_S | x^*_{i,S})}{(\sum_{k=1}^L d(x_S | x^*_{k,S}))^2} \right\}_{i=1}^L$.

Since we do not know $p(\hat{\phi})$ to use the results from Supplement H, we instead approximate the variance of the smoothed Shapley values by

$$\mathrm{Var}[\hat{\phi}^{smoothed}(j, x)] = \frac{1}{(\sum_{i=1}^L w_i)^2} \sum_{S \subseteq \{1,\ldots,m\}} w_i^2 \sigma^2_{\hat{\phi}_i}$$

which follows again by independence and $\mathrm{Var}[aX] = a^2 \mathrm{Var}[X]$ where $\sigma^2_{\hat{\phi}_i}$ is the variance of the estimator $\hat{\phi}(j, x_i)$.

| Data Set | # Observations | # Features | URL |
|---|---|---|---|
| adult | 32.561 | 11 | https://archive.ics.uci.edu/ml/datasets/adult |
| bike | 17.389 | 15 | https://archive.ics.uci.edu/ml/datasets/bike+sharing+dataset |
| Boston | 506 | 14 | https://scikit-learn.org/stable/modules/generated/sklearn.datasets.load_boston.html |
| compas | 6.172 | 12 | https://github.com/dylan-slack/Fooling-LIME-SHAP/tree/master/data |
| mnist | 60.000 | 784 | https://pytorch.org/vision/stable/datasets.html |

Table 3: Details on all real-world data sets used in the experiments.

| Experiment | Type of Compute | Amount of Compute (in clock-time seconds) |
|---|---|---|
| Simulated experiment | 2.4 GHz 6-Core Intel Core i5-9300H CPU | 177 |
| Adversarial attack | 2.6 GHz 4-Core Intel Xeon Gold 6240 CPU | 4,377 |
| Lipschitz continuity | 2.4 GHz 6-Core Intel Core i5-9300H CPU | $\approx 14,400$ (correct up to hour) |
| adult | 2,6 GHz 6-Core Intel Core i7 | 2,968 |
| Boston | 2,6 GHz 6-Core Intel Core i7 | 2,741 |
| bike | 2,6 GHz 6-Core Intel Core i7 | 427 |

Table 4: Total amount of compute for selected experiments.

# K  Experimental Details and Additional Experimental Results

In this section, we provide additional details on the experiments as presented in the main part, present complete results and add additional experiments which illustrate how our approaches perform.

## K.1  Experimental Details

For our experiments, we used the code from the following public resources:

- KernelSHAP [6], available at `https://github.com/slundberg/shap` with MIT License

- Explanation GAME [7], available at `https://github.com/fiddler-labs/the-explanation-game-supplemental` (no license)

- LIME [11], available at `https://github.com/marcotcr/lime` with MIT License

- Fooling-LIME-SHAP [15], available at `https://github.com/dylan-slack/Fooling-LIME-SHAP` with MIT License

- MeLIME [2], available at `https://github.com/tiagobotari/melime` with MIT License

- Robust-interpret [1], available at `https://github.com/dmelis/robust_interpret` (no license)

Black box models and other classification algorithms were trained using `sklearn` with default parameters. Please refer to Table 3 for details on the data sets. Data was splitted with a 80/20 split into train and test data randomly given the seeds, unless otherwise specified. All models were trained after data scaling with a standard scaler. We used the euclidean distance as a distance measure for continuous features. For categorical data, distance was set to one if the reference point had a different feature value, 0 otherwise. We used an exponential kernel as distance distribution $d(x^*|x)$. Following [7], we limited the visual inspection of the SHAP attributions on the UCI data sets presented in the main part of the paper in Figure 6 to the top 5 features which were selected using the Light GBM "split" feature importance, computed from the numbers of times the feature is used in a model.

## K.2  Simulated Experiments

Simulated experiments comparing Neighbourhood SHAP and standard SHAP are included and explained in Figure 6. The local linear approximation presented in the main part and in Supplement A were computed with a weighted Ridge Regression using the `LimeBASE` function of the LIME package. The data was sampled from the corresponding background data sets and the weights were computed with euclidean distance based exponential kernels.

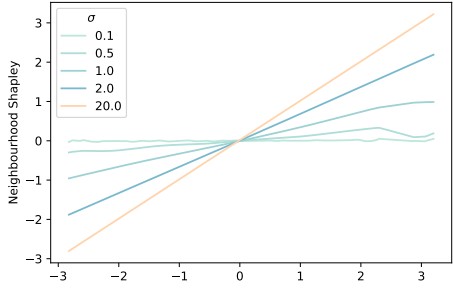
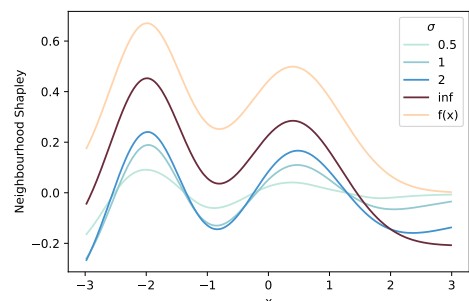

(a) For $\sigma = 20$, the Neighbourhood Shapley values of $f(x) = x$ are visually not distinguishable from Shapley values. The smaller $\sigma$, the smaller the Shapley values since the expected model outcome within the neighbourhood of $x$ gets closer to $f(x)$. Further, local Shapley values with small bandwidths are instable in the boundary areas where data is sparse.

(b) Here the black box function is the CDF of the balanced mixture of $\mathrm{Normal}(-2, 0.6)$ and $\mathrm{Normal}(0.4, 1)$. The Shapley value attribution at $x = -1$ is positive, even though increasing $x$ in a small neighbourhood would lead to a decrease in $f(x)$. The Neighbourhood Shapley values reflect this behaviour.

Figure 6: Neighbourhood Shapley values for 1000 different $x$ sampled from $\mathrm{Normal}(0, 1)$ and different bandwidths $\sigma$.

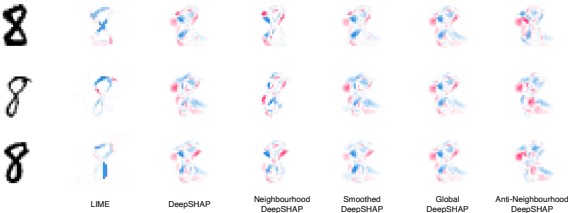
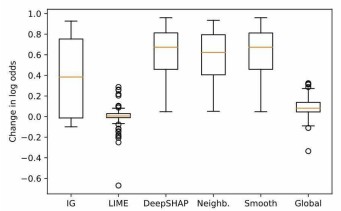

(a) Attributions for the digit '3' of images trained on 100 background samples consisting of images of a 3 or an 8. We used 100 reference points and a bandwidth of $0.1 \cdot 784$ for the DeepSHAP alternatives. The Quickshift segmentation algorithm with a kernel size of 1, maximal distance of 1 and ratio of 0.95 was trained to obtain the highest possible number of segments (here 28) for the LIME algorithm.

(b) Change in Log odds when masking 100 randomly sampled images of an 8 to explain a 3, with a background data set of 100. We obtained a bandwidth of $0.25 \cdot 784$ for the Neighbourhood DeepSHAP approach and of $0.1 \cdot 784$ for the Smoothed DeepSHAP on a validation data set by optimising the bandwidth over a grid of $[0.05, 0.1, ... 0.25, 0.3]$.

Figure 7: Predictive results on the MNIST data set.

### K.3 Image Classification

We applied the Neighbourhood and the Smoothed Shapley values also on the MNIST data set. We trained a convolutional network as in `https://github.com/slundberg/shap/blob/master/notebooks/image_examples/image_classification/PyTorch%20Deep%20Explainer%20MNIST%20example.ipynb` for ten epochs. We then applied DeepSHAP, Neighbourhood DeepSHAP, Smoothed DeepSHAP, Anti-Neighbourhood DeepSHAP (difference between DeepSHAP and Neighbourhood DeepSHAP). Refer to Figure 7 for the results.

In a second step, we mimicked the experimental set-up of [1] with results presented in Figure 8. We see that even though small amount of smoothing does not change the visual representation of the explanation in a detectable manner (Figure 7b), the Lipschitz estimates introduced by [1] decrease considerably.

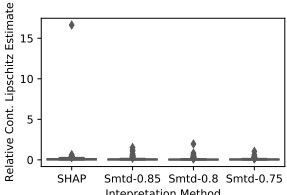
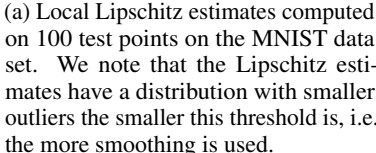
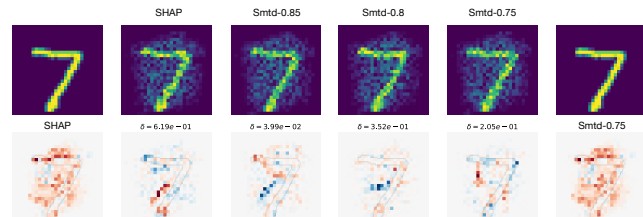

(a) Local Lipschitz estimates computed on 100 test points on the MNIST data set. We note that the Lipschitz estimates have a distribution with smaller outliers the smaller this threshold is, i.e. the more smoothing is used.

(b) Explanations of a CNN model prediction's on an example MNIST digit with Gaussian noise added to it. Here $\delta$ is the ratio $\|f(x) - f(x')\|_2/\|x - x'\|_2$ for the perturbed $x$ which was chosen such that it maximises the Lipschitz estimate of the explanation model on the test instance. The original digit image $x'$ has been once explained with DeepSHAP and once with Smoothed DeepSHAP. We see that $\delta$ decreases with the amount of smoothing.

Figure 8: Results on Lipschitz continuity of SHAP and Smoothed SHAP on the MNIST data set. Smtd-0.8 denotes the Smoothed SHAP estimate where the bandwidth was chosen as large as possible such that the largest normalised weight is smaller than 0.8.

## K.4 Adult Data

On the adult data we trained a LightGBM Classifier trained with `sklearn` with default parameters to predict an annual income of more than \$50,000. When analysing the data set we consider an individual with the following features

- `capital gain` = 0

- `age` = 43

- `relationship` = Not-in-family

- `education number` = 12

- `marital-status` = Never-married

- `annual-income above` \$50,000 = True

- and a black box prediction of 0.094

The training set had 32561 instances whilst the test set had 16281 instances. Observations with missing values were deleted (N=3,620). The model fit was measured with the accuracy $\alpha$. $\alpha = 14.3\%$ on the train set and $\alpha = 14.7\%$ on the test set for the parsimonious model. $\alpha = 11.0\%$ on the train set and $\alpha = 12.7\%$ on the test set for the entire model.

Following experimental results have been included in the following

- Descriptive plots for explaining attributions away from the centre of mass in Figure 9

- Bias and variance when using Smoothed SHAP in Figure 12a (please see Figure 27 for a plot of the bias of Smoothed SHAP over multiple tabular data sets.)

- A bar plot for comparing standard Shapley with Smoothed SHAP in Figure 12b

- Box plots to compare the variance across methods in Figure 10

- Plots for varying kernel widths for the parsimonious model in Figure 11

- Neighbourhood SHAP for varying kernel widths: Entire model in Figure 13

- Smoothed SHAP results across background data set in Figure 14

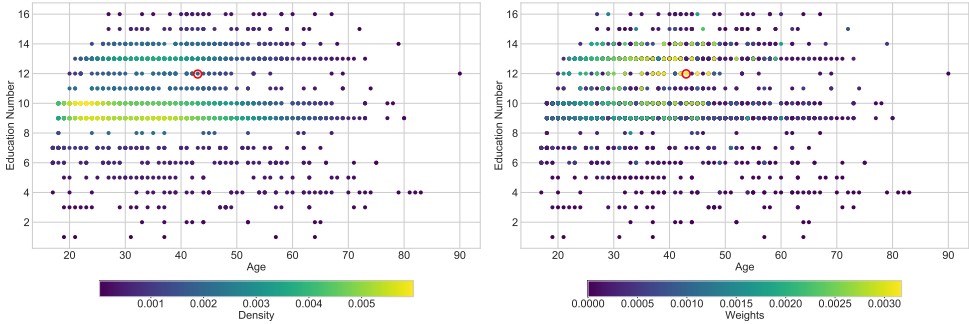

(a) Coloured by kernel-density estimates using Gaussian kernels.

(b) Coloured by Neighbourhood SHAP weights for the individual of interest. Kernel width was equal to 2.

Figure 9: Scatter plot of the Education Number by Age across reference points. Individual of interest lies in the red circle.

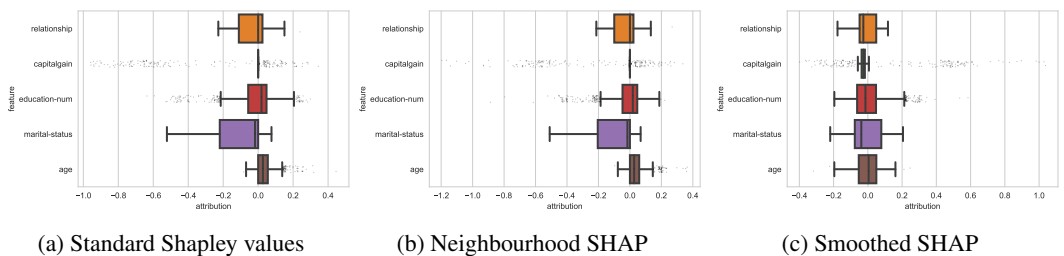

(a) Standard Shapley values      (b) Neighbourhood SHAP      (c) Smoothed SHAP

Figure 10: Comparison of box plots on the parsimonious model: standard Shapley values, Neighbourhood SHAP, Smoothed SHAP. Kernel width equal to 5.

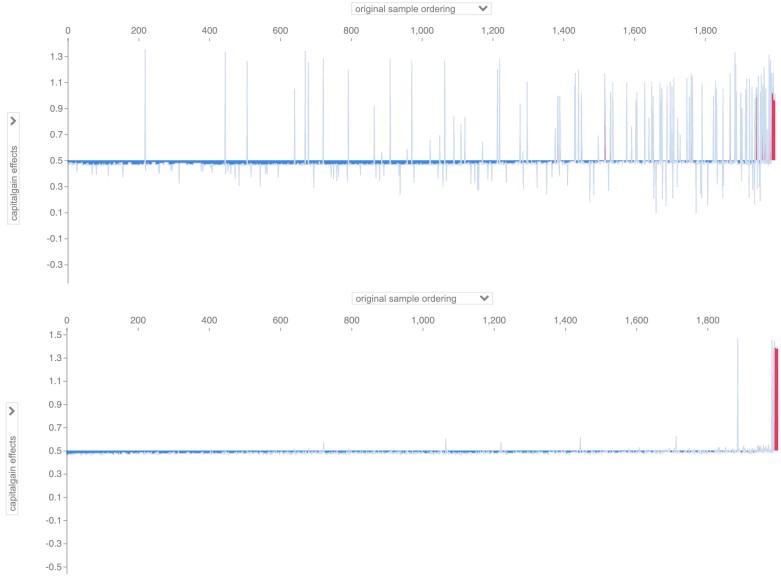

Figure 14: Comparison of the marginal Shapley values (top) and Smoothed SHAP with kernel width 5 (bottom) for computing the effects of Relationship in the Adult Income data set using the parsimonious model. Feature attributions are sorted by similarity according to a preliminary PCA analysis across a subset of 2000 samples from the Adult Income data set, using 2000 reference points. Plot was created using the KernelSHAP package [6].

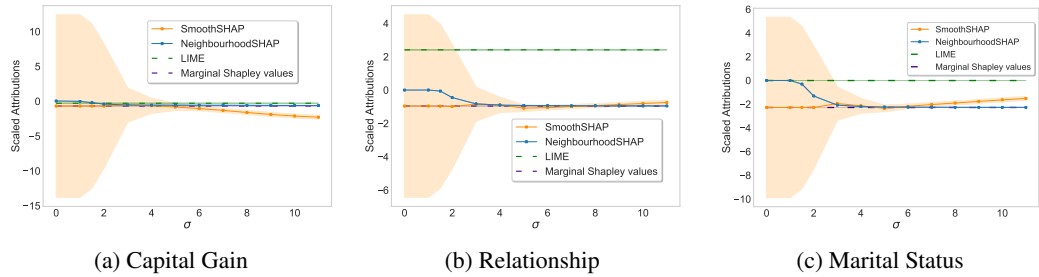

| (a) Capital Gain | (b) Relationship | (c) Marital Status |

Figure 11: Scaled attributions at a given test instance for varying kernel widths computed with 2000 reference points in the adult data sets. Bounds for LIME have been computed over 2000 runs, while the Shapley bounds have been estimated with their theoretical formula as outlined in Supplement J.

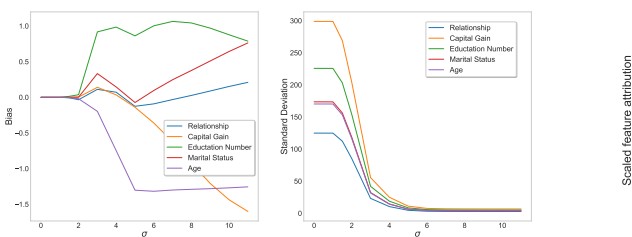 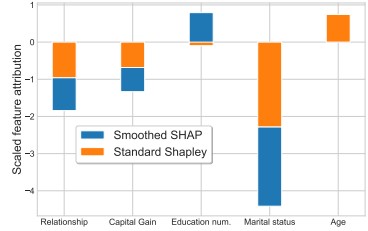

(a) Bias and standard deviation for Smoothed SHAP values of our individual of interest by kernel widths, using 2000 reference points.

(b) Comparison of standard Shapley values and Smoothed SHAP with a kernel width equal to 2, using 2000 reference points.

Figure 12

## K.5 Bike Data

On the bike data we trained a LightGBM Regressor trained with `sklearn` with default parameters to predict the hourly number of bike rentals. The hour we analyse in the bike data is

- `workingday` = True
- `hour` = 10
- `temperature` = 0.82
- `humidity` = 0.56
- `season` = 3
- `hourly number of bike rentals` = 218
- and prediction of 110.7

The training set had 8645 instances whilst the test set had 8734 instances. The model fit was measured with the coefficient of determination. $R^2 = 0.934$ on the train set and $R^2 = 0.632$ on the test set for the parsimonious model. $R^2 = 0.959$ on the train set and $R^2 = 0.641$ on the test set for the entire model. Similarly to before we have included following results for the bike data set:

- Descriptive plots for explaining attributions away from the center of mass in Figure 15
- Bias and variance when using Smoothed SHAP in Figure 18a
- A bar plot for comparing standard Shapley with SmoothedSHAP in Figure 18b
- Box plots to compare the variance across methods in Figure 16
- Plots for varying kernel widths for the parsimonious model in Figure 17
- Neighbourhood SHAP for varying kernel widths: Entire model in Figure 20
- Smoothed SHAP results across background data set in Figure 19

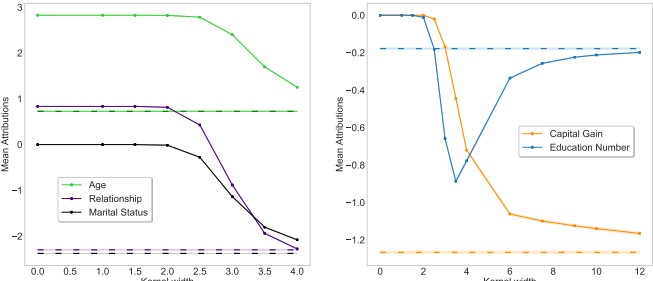

Figure 13: Mean attributions of Neighbourhood for varying kernel widths, with our model trained on all 14 features. Only features from the parsimonious model are represented in this plot. Values are computed for our individual of interest, using 2000 reference points. Marginal Shapley values correspond to dashed lines; Neighbourhood SHAP values correspond to solid lines.

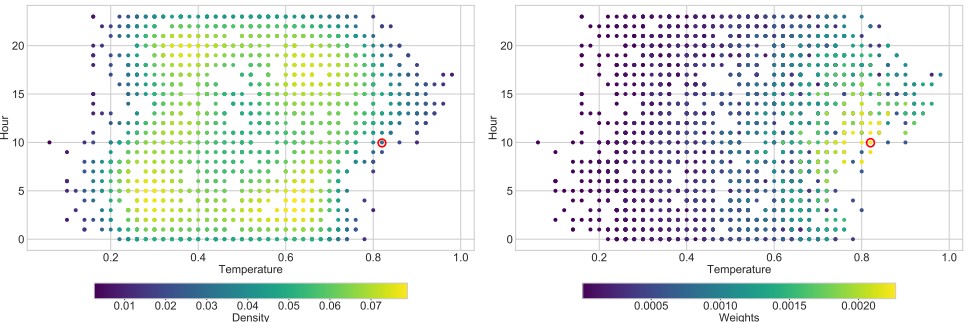

(a) Coloured by kernel-density estimates using Gaussian kernels.

(b) Coloured by Neighbourhood SHAP weights for the individual of interest. Kernel width was equal to 2.

Figure 15: Scatter plot of `hour` by `Temperature` across reference points. Individual of interest lies in the red circle.

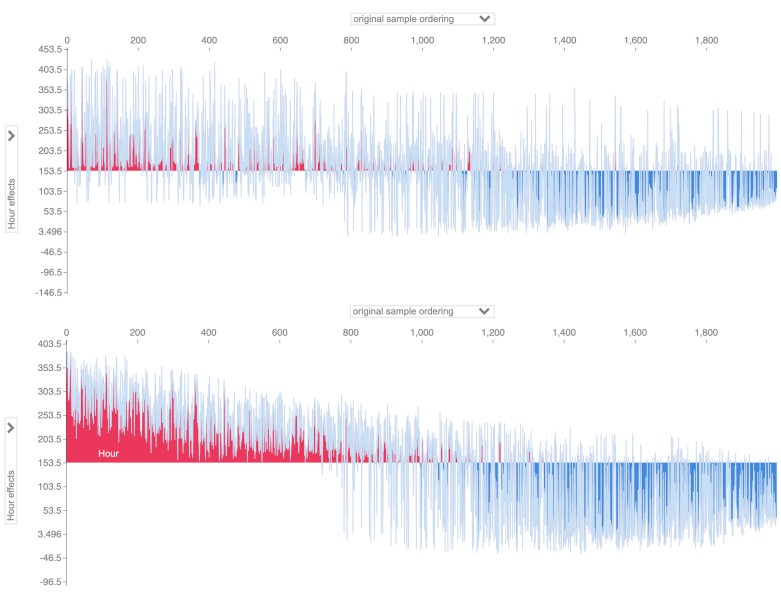

Figure 19: Comparison of the marginal Shapley values (top) and Smoothed SHAP with kernel width 2 (bottom) for computing the effects of `hour` in the bike data set using the parsimonious model. Feature attributions are sorted by similarity according to a preliminary PCA analysis across a subset of 2000 samples from the Adult Income data set, using 2000 reference points. Plot was created using the KernelSHAP package [6].

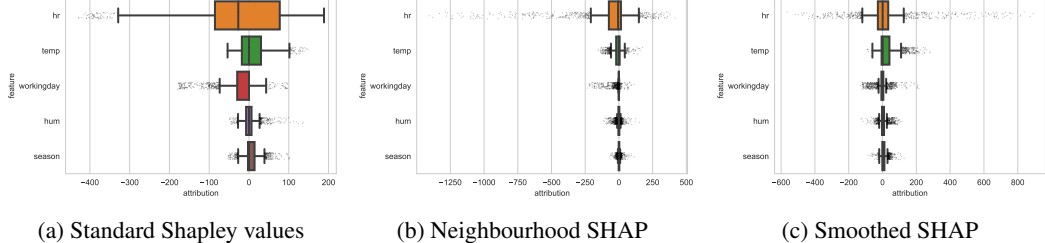

| (a) Standard Shapley values | (b) Neighbourhood SHAP | (c) Smoothed SHAP |
|---|---|---|

Figure 16: Comparison of box plots on the parsimonious model: standard Shapley values, Neighbourhood SHAP, Smoothed SHAP. Kernel width equal to 2.

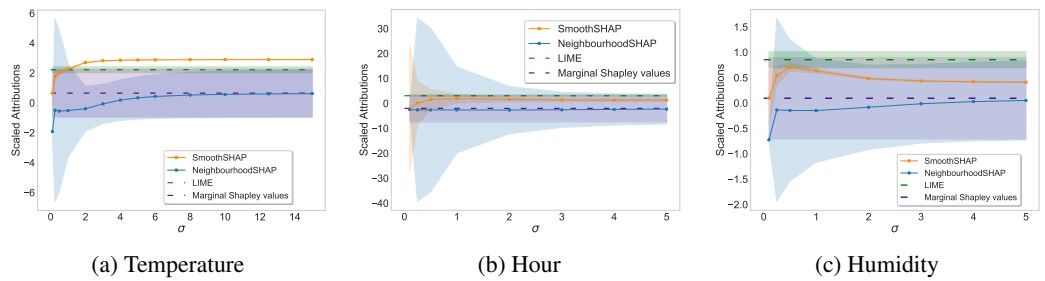

| (a) Temperature | (b) Hour | (c) Humidity |
|---|---|---|

Figure 17: Scaled attributions at a given test instance for varying kernel widths computed with 2000 reference points in the adult data sets. Bounds for LIME have been computed over 2000 runs, while the Shapley bounds have been estimated with their theoretical formula as outlined in Supplement J.

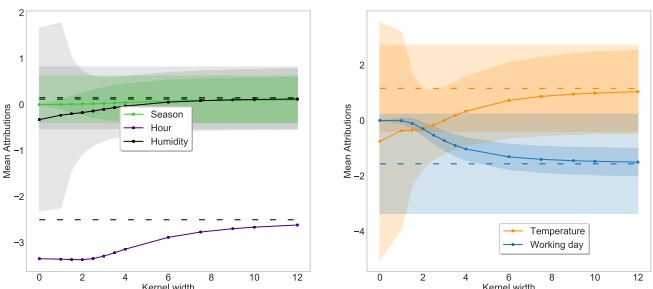

Figure 20: Mean attributions of Neighbourhood SHAP for varying kernel widths, with our model trained on all 10 features. Only features from the parsimonious model are represented in this plot. Values are computed for the individual of interest, using 2000 reference points. Marginal Shapley values correspond to dashed lines; Neighbourhood SHAP values correspond to solid lines. For `hour`, the standard error is not represented as it is too wide: approximately 128 for the standard Shapley value, while it decreases from approximately 582 std to 131 for the Neighbourhood Shapley values.

## K.6 Boston Housing Data

On the Boston housing data we trained a LightGBM Regressor with `sklearn` with default parameters to predict the median value of owner-occupied homes in \$1000's. The dwelling analysed has following properties

- Percentage of lower status of the population (LSTAT) = 18.8
- Average number of rooms per dwelling (RM) = 6.05
- Per capita crime rate by town (CRIM) = 5.29
- Weighted distances to five Boston employment centres (DIS) = 2.17
- Pupil-teacher ratio by town (PTRATIO) = 20.2
- Median value in \$1000's = 23.2
- and prediction of 16.4

The training set had 404 instances whilst the test set had 102 instances. The model fit was measured with the coefficient of determination. $R^2 = 0.963$ on the train set and $R^2 = 0.661$ on the test set for

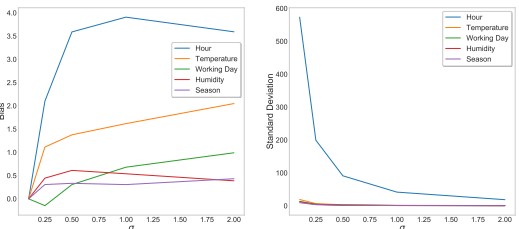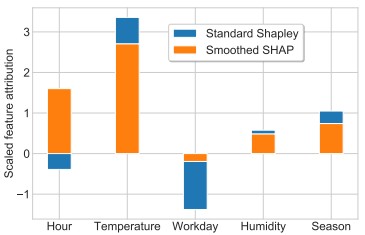

(a) Bias and standard deviation for Smoothed SHAP values of our individual of interest by kernel widths, using 2000 reference points.

(b) Comparison of standard Shapley values and Smoothed SHAP with a kernel width equal to 2, using 2000 reference points.

Figure 18

the parsimonious model. $R^2 = 0.977$ on the train set and $R^2 = 0.699$ on the test set for the entire model.

- Descriptive plots for explaining attributions away from the center of mass in Figure 21
- Bias and variance when using Smoothed SHAP in Figure 24a
- A bar plot for comparing standard Shapley with SmoothedSHAP in Figure 24b
- Box plots to compare the variance across methods in Figure 22
- Plots for varying kernel widths for the parsimonious model in Figure 23
- Neighbourhood SHAP for varying kernel widths: Entire model in Figure 25
- Smoothed SHAP results across background data set in Figure 26

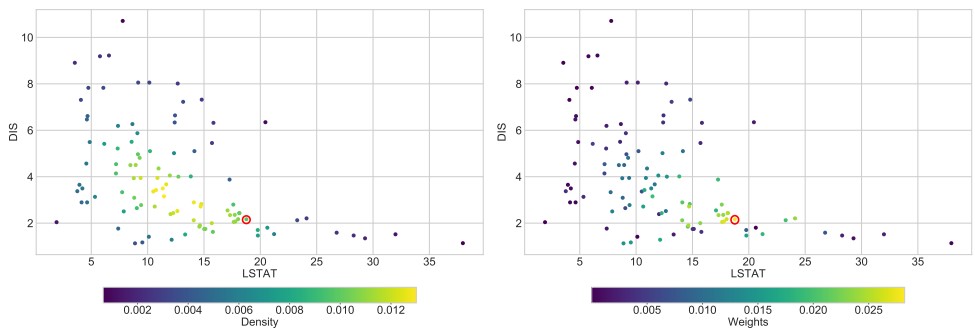

(a) Coloured by kernel-density estimates using Gaussian kernels.

(b) Coloured by Neighbourhood SHAP weights for the individual of interest. Kernel width was equal to 2.

Figure 21: Scatter plot of the Percentage of lower status population (LSTAT) by Weighted distances to five Boston employment centres (DIS) across reference points. Individual of interest lies in the red circle.

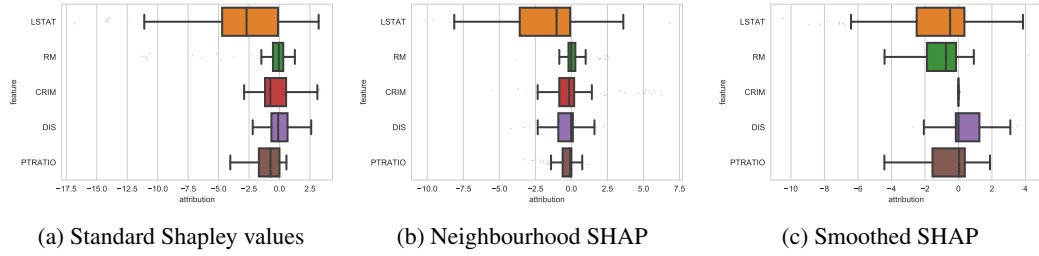

(a) Standard Shapley values

(b) Neighbourhood SHAP

(c) Smoothed SHAP

Figure 22: Comparison of box plots on the parsimonious model: standard Shapley values, Neighbourhood SHAP, Smoothed SHAP. Kernel width equal to 2.

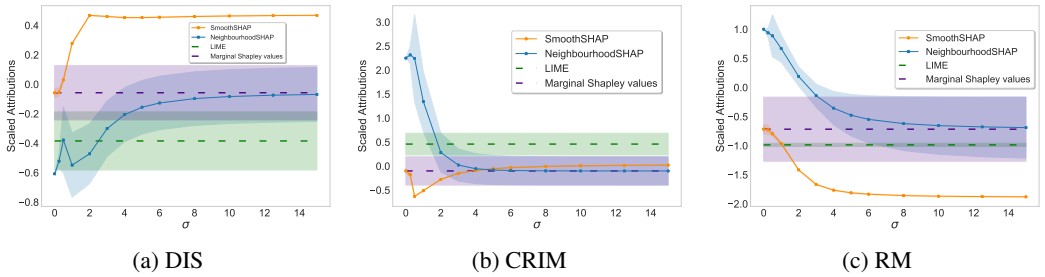

(a) DIS          (b) CRIM          (c) RM

Figure 23: Scaled attributions at a given test instance for varying kernel widths computed with 102 reference points in the adult data sets. Bounds for LIME have been computed over 2000 runs, while the Shapley bounds have been estimated with their theoretical formula as outlined in Supplement J.

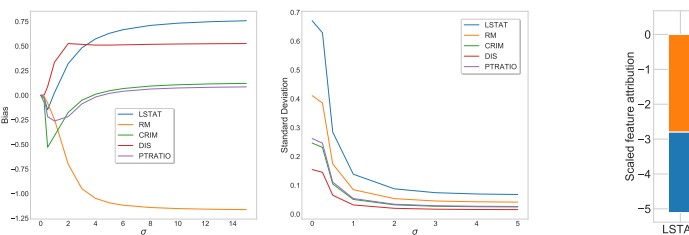
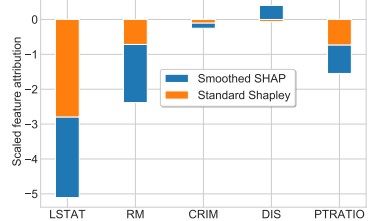

(a) Bias and standard deviation for Smoothed SHAP values of our individual of interest by kernel widths, using 102 reference points.

(b) Comparison of standard Shapley values and Smoothed SHAP with a kernel width equal to 2, using 102 reference points.

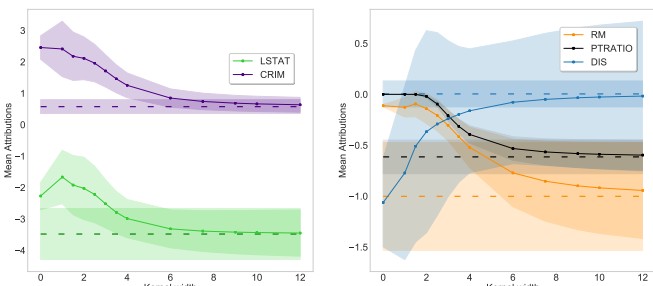

Figure 25: Mean attributions of Neighbourhood SHAP for varying kernel widths, with our model trained on all 10 features. Only features from the parsimonious model are represented in this plot. Values are computed for our individual of interest, using 102 reference points. Marginal Shapley values correspond to dashed lines; Neighbourhood SHAP values correspond to solid lines.

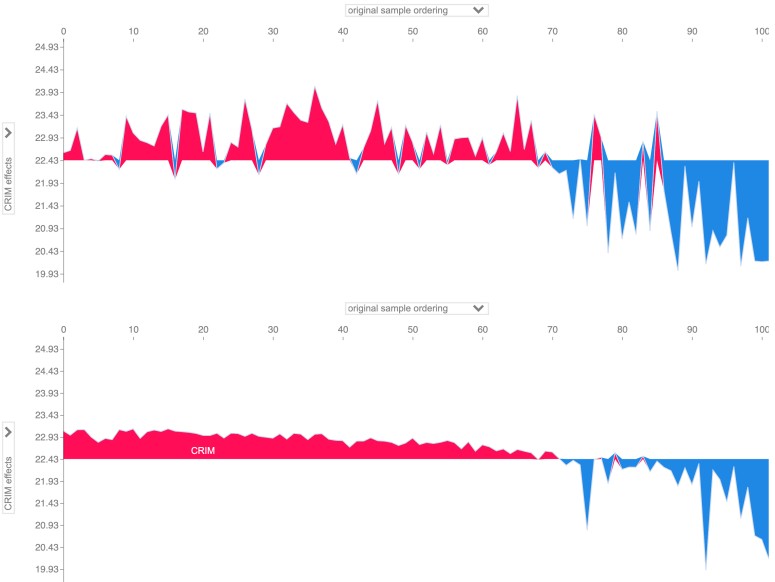

Figure 26: Comparison of the marginal Shapley values (top) and Smoothed SHAP with kernel width 2 (bottom) for computing the effects of CRIM in the bike data set using the parsimonious model. Feature attributions are sorted by similarity according to a preliminary PCA analysis across a subset of 102 samples from the Adult Income data set, using 2000 reference points. Plot was created using the KernelSHAP package [6].

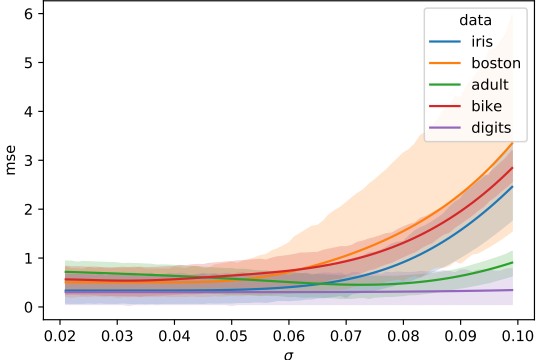

Figure 27: Bias of Smoothed SHAP for different kernel widths. While most data sets reach their minimum MSE at a positive kernel width (bike: 0.03, adult: 0.08, Boston: 0.03, digits: 0.1, iris: 0), this minimum is insignificant.

### K.7 ROAR metric

|  | Adult ($\alpha$) | | Bike ($R^2$) | | Boston ($R^2$) | |
|---|---|---|---|---|---|---|
|  | Local mean | Global mean | Local mean | Global mean | Local mean | Global mean |
| **SHAP** | 0.276 | 0.098 | 0.232 | 0.189 | 0.349 | 0.154 |
| **Neighbourhood SHAP** | 0.364 | 0.271 | 0.259 | 0.312 | 0.389 | 0.320 |
| **Smoothed SHAP** | 0.233 | 0.280 | 0.391 | 0.306 | 0.177 | 0.399 |

Table 5: Absolute change in evaluation metrics ($R^2$ or accuracy) on the test data when imputing with local and global mean. Abbreviations: $\alpha$=accuracy