# OpenReview forum: "On Locality of Local Explanation Models"
_NeurIPS.cc/2021/Conference — NeurIPS 2021 Poster_

### Official Review · Reviewer_iHps · 2021-07-15

**Rating:** 7
**Confidence:** 3

**Summary:**

The main contributions of this paper are two variants of SHAP explanation method, Neighborhood SHAP and Smoothed SHAP. These two methods are motivated from the prior work in extending SHAP only captures the global behavior of the model but leaves the local behavior unexplored. The author demonstrates that the proposed explanations, neighborhood SHAP and Smoothed SHAP provide better local explanations over other SHAP methods in several data distributions.

**Limitations And Societal Impact:**

The authors have adequately addressed the limitations and potential negative societal impact of their work

**Main Review:**

This is an interesting and complete work in general. The authors have provided detailed theoretical analysis in extending the SHAP method with a distribution that is more useful to capture the model’s local behavior. The technical part is sound. My main questions are about the motivation of Smoothed SHAP and some analysis in the experiment section.

### Originality
The paper extends SHAP explantion with distributions that can capture the model’s local behaviors, which is the motivation for Neighborhood SHAP. However, the transition to Smoothed SHAP seems to be a little sudden such that even without Smoothed SHAP, Neighborhood SHAP seems to stand on its own. The only relation explained in the paper is they are related to the same technology, Nadaraya-Watson estimator, in different ways (line 161-162). It would be interesting to discuss how the way of treating neighborhood samplers as Nadaraya-Waston estimator provides the reader a better understanding behind Smoothed SHAP and Neighborhood SHAP. For example, is there some unified view of these two approaches?

Other than that, the related work is well-cited and the novelty of this paper is significant.

### Quality
The theoretical part is sound and the authors have provided very useful appendix content for the readers. I have several questions about the analysis of Neighborhood SHAP and Smoothed SHAP.

- How do we choose between Neighborhood SHAP and Smoothed SHAP? When analyzing these two approaches at page 8, the baseline approach is mostly the previous SHAP methods, e.g. Marginal SHAP for both methods, respectively. But what should the human user trust more when represented with Neighborhood SHAP and Smoothed SHAP as there are still obvious gap between the results shown in Figure 6. When the relation between Neighborhood SHAP and Smoothed SHAP is not well-described in the previous section, it would be very useful to discuss which method is better in terms of local explainability and what is the suggested way in this paper to interpret result of the Neighborhood SHAP against Smoothed SHAP in an explicit way.

- Baseline choices in deep models. Comparing the neighborhood version and smoothed version of SHAP with LIME are reasonable choices for tree models or any models that are treated as black boxes. However, in the paragraph that shows Neighborhood DeepSHAP is more insightful than LIME seems to be less fair since DeepSHAP is based on DeepLIFT which has access to the model’s architecture and backpropagation while LIME is model-agnostic and therefore may not able to be faithfully capture the deep model’s behavior. The comparisons may be more fair if the baselines are also leveraging the gradient information, e.g. Integrated Gradient [1] and Smoothed Gradient [2].

### Clarity
The paper is well-organized. The writing is easy to follow even though there are some typos. The analysis in the experimental section is quite dense and there might be some ways to improve these paragraphs to make it easier to read.
### Significance
This paper’s result is significant. Other researchers and especially the users of machine learning products may get benefit from the work.


[1] Sundararajan, M., Taly, A., & Yan, Q. (2017). Axiomatic Attribution for Deep Networks. ICML 2017
[2] Smilkov, D., Thorat, N., Kim, B., Viégas, F.B., & Wattenberg, M. (2017). SmoothGrad: removing noise by adding noise. ArXiv, abs/1706.03825.


**Time Spent Reviewing:**

9

---

> ### Author Response · Authors · 2021-08-10
> **We thank Reviewer iHps for appreciating the significance of the work. A comparison with Integrated Gradients has been added to a revised version of the paper.**
>
> **"How do we choose between Neighborhood SHAP and Smoothed SHAP? When analyzing these two approaches at page 8, the baseline approach is mostly the previous SHAP methods, e.g. Marginal SHAP for both methods, respectively. But what should the human user trust more when represented with Neighborhood SHAP and Smoothed SHAP as there are still obvious gap between the results shown in Figure 6. When the relation between Neighborhood SHAP and Smoothed SHAP is not well-described in the previous section, it would be very useful to discuss which method is better in terms of local explainability and what is the suggested way in this paper to interpret result of the Neighborhood SHAP against Smoothed SHAP in an explicit way."**
>
> The choice of Smoothed SHAP / Neighborhood SHAP depends on the kind of interpretation the user is aiming for. As noted, Neighbourhood SHAP is a stand-alone method. Its values compute feature attributions when contrasting with model behaviour at similar observations. For example, a person might be interested to understand why the model predicts high income for them when contrasting to their peers. Smoothed SHAP was listed as a complement of Neighbourhood SHAP: while Neighbourhood SHAP models feature **removal** with local distributions, Smoothed SHAP models feature **inclusion** with local distributions. As a result, Smoothed SHAP summarises marginal Shapley values which contrast against the entire population within a neighbourhood, and not a single instance. For instance, one might use Smoothed SHAP to understand how feature attributions vary within strata of the population, for a given model. It is thus a step towards global feature attribution. If there is a drastic change in Smoothed SHAP in small neighbourhoods we know that the SHAP values of the neighbours are quite different from the SHAP values of the test instance.
>
> **"Baseline choices in deep models. Comparing the neighborhood version and smoothed version of SHAP with LIME are reasonable choices for tree models or any models that are treated as black boxes. However, in the paragraph that shows Neighborhood DeepSHAP is more insightful than LIME seems to be less fair since DeepSHAP is based on DeepLIFT which has access to the model’s architecture and backpropagation while LIME is model-agnostic and therefore may not able to be faithfully capture the deep model’s behavior. The comparisons may be more fair if the baselines are also leveraging the gradient information, e.g. Integrated Gradient and Smoothed Gradient."**
>
> We also appreciate the comment on the fairness of the comparison on the MNIST data set. LIME was included as a baseline since it is often used for model explainability on images [1, 2], and it does leverage image segmentation which the gradient methods do not rely on. Since the reviewer still makes a valid point, results from Integrated Gradients (IG) were added to Figures 1 (motivating example) and 7 (MNIST experiment) in a revised version of the paper.
>
> IG, in particular, resembles feature-removal based approaches in so far as it requires baseline values to be specified. This baseline is typically defined to be just the mean of the feature. If a fixed baseline value is chosen, IG suffers under the same globality problem that was outlined for SHAP (Figure 1). Further, we see that the attributions of IG are quite similar to those of LIME in Figure 2. Figure 3 shows that masking with IG leads to the second-lowest average change in log odds and it thus underperforms (insignificantly) compared to the SHAP alternatives. A detailed comparison to Neighbourhood SHAP (i.e. black-box access vs white-box access, neighbourhood reference distribution vs baseline value) has been added to the Supplementary Material.
>
> [anonymous URL that we can share if needed]
> (Figure 1:) Feature attributions for motivating example.
>
> [anonymous URL that we can share if needed]
> (Figure 2:) Feature attributions on randomly picked MNIST images.
>
> [anonymous URL that we can share if needed]
> (Figure 3:) Change in log odds. See Appendix K.2 or [3] for a description of the experiment.
>
>
> ### References
>
> [1] Alvarez-Melis, D. and Jaakkola, T. S. (2018). On the robustness of interpretability methods. arXiv preprint arXiv:1806.08049.
>
> [2] Botari, T., Hvilshøj, F., Izbicki, R., and de Carvalho, A. C. (2020). Melime: Meaningful local explanation for machine learning models. arXiv preprint arXiv:2009.05818.
>
> [3] Lundberg, S. and Lee, S.-I. (2017). A unified approach to interpreting model predictions. arXiv preprint arXiv:1705.07874.

---

> > ### Comment · Reviewer_iHps · 2021-08-11
> > **A Follow-up question for Smoothed SHAP**
> >
> > I appreciate the authors' thoughtful response and my questions on baseline comparisons are addressed.
> >
> > However, as reading the discussion with reviewer RMYP and the corresponding responses for my question, I realize the authors have tried to emphasize the fact that NeighborhoodSHAP is a stand-alone method that should be able to fully answer the question the authors ask in the first place as said in the contribution part *We propose Neighbourhood SHAP (Section 3) which considers local reference populations
> > 36 for prediction points as a complimentary approach to SHAP.* Therefore, I start to have a mixed feeling about **what is the necessity of including SmoothedSHAP together with NeighborhoodSHAP** in this paper. The authors have presented a lot more comparisons, i.e. faithfulness, stability and what kind of questions they can answer, i.e. "The choice of Smoothed SHAP / Neighborhood SHAP depends on the kind of interpretation the user is aiming for" . As more information is provided in the authors' feedback, I see that SmoothedSHAP is more like a parallel method to NeighborhoodSHAP because they are clearly solving different questions. An analogy of this result is I was presented with a paper which includes the Integrated Gradient and Smoothed Gradient all together. Even with two methods being technical sound and useful, a mixture of NeighborhoodSHAP and SmoothedSHAP loses a tight bound in-between such that it is currently not intuitive to expect SmoothedSHAP as a natural follow-up of the NeighborhoodSHAP we would like to discuss.
> >
> > Maybe I am wrong but please help me to understand **what is the necessity of including SmoothedSHAP together with NeighborhoodSHAP such that without SmoothedSHAP the other method is not able to address the question proposed by this paper in the very beginning?**

---

> > > ### Author Response · Authors · 2021-08-16
> > > **Smoothed SHAP and Neighbourhood SHAP are complementary approaches that generalize the locality of local explanation models. We believe a joint analysis including both methods provides a well-rounded contribution.**
> > >
> > > We thank Reviewer iHps for acknowledging the efforts made in our previous comment, and for their swift response. As mentioned in the response, Smoothed SHAP and Neighbourhood SHAP are indeed two parallel methods. However, they both fit into the overarching theme of the paper namely, exploring (and improving) the locality of local explanation methods.
> > >
> > > On the one hand, Neighbourhood SHAP shows how modelling feature removal with neighbourhood distributions improves the locality of Shapley values. On the other hand, Smoothed SHAP shows how modelling feature inclusion with neighbourhood distributions improves the stability of SHAP and can highlight local variability. Tabular LIME presents an alternative method to Smoothed SHAP, the former dividing the dimension of each feature space into different quantiles (see lines 207-210 and Appendix H), whereas Smoothed SHAP uses a neighbourhood distribution that is centred on the observation of interest. SmoothedSHAP thus fits into the overarching story of generalising the locality of local explanation models and addresses a locality issue in a more targeted fashion than Tabular LIME.
> > >
> > > Smoothed SHAP and Neighbourhood share the same underlying technique (i.e. modelling feature inclusion/removal respectively with neighbourhood distributions), and we believe it's valuable to consider a joint exposition that allows for comparisons. The current submission includes comparisons on the interpretability and stability between these two methods. Such a detailed discussion is only possible within the same piece of work.

---

### Official Review · Reviewer_74h4 · 2021-07-16

**Rating:** 7
**Confidence:** 3

**Summary:**

The authors highlight on some of the flaws of SHAP for local explanations and propose a new variant, Neighbourhood SHAP, based on neighbourhood sampling and a kernel regressor.

Authors do a detailed theoretical and experimental analysis of the effects of the hyperparameters (i.g. kernel bandwith) of this technique and how they can be interpreted.
They consider several tabular datasets and one example on MNIST. They compare their insights with other common techniques (i.g. kernelSHAP).

Overall, authors further the theoretical results of interpretability tools.

**Limitations And Societal Impact:**

Authors did not address it, they should be able to do it.

**Main Review:**

Work is original, clear, and very extensive. Furthermore is provides theoretical insights to a relevant method in XAI. I think this work should be accepted and would improve with some minor edits.

Some points that authors should address and would make me consider to slightly improve their score (8 or 9):

* The title is pretty broad "On Locality of Local Explanation Models", when it actually only addressing a class of local explanation models ("SHAP").
* Authors might want to expand and contextualize their results within the broader family of XAI techniques like attribution/saliency techniques ( CAM/GradCAM, integrated gradients) or Layer-wise Relevance Propagation (LRP), especially since there are connections between these techniques and Shapley values [1].
* The MNIST example is mostly anecdotal, stronger results would be quantitative. There is already a wave of works that are attempting to evaluate the quality of explanations [2-7]. Adopting some of these ideas/frameworks would give stronger evidence for claims.

Authors should write their Societal Impact section.

Citations:
[1] https://arxiv.org/abs/1903.10992
[2] https://papers.nips.cc/paper/2020/hash/417fbbf2e9d5a28a855a11894b2e795a-Abstract.html
[3] https://arxiv.org/abs/1907.09701
[4] https://arxiv.org/abs/1904.11829
[5] https://arxiv.org/abs/2003.07258
[6] https://arxiv.org/abs/1806.10758
[7] https://arxiv.org/abs/1806.10758

**Time Spent Reviewing:**

2 hrs

---

> ### Author Response · Authors · 2021-08-10
> **We thank Reviewer 74h4 for appreciating the originality of our work. A comparison with Integrated Gradients and additional quantitative evaluation metrics have been added to a revised version of the paper, and the title has been changed.**
>
> **"The title is pretty broad "On Locality of Local Explanation Models", when it actually only addressing a class of local explanation models ("SHAP")."**
>
> Thank you for the suggestion. Since our ideas generalise to all removal-based local explainability methods (i.e. we explicitly mention LIME), we changed the title now to "On Locality of Removal-Based Local Explanation Models".
>
> **"Authors might want to expand and contextualize their results within the broader family of XAI techniques like attribution/saliency techniques ( CAM/GradCAM, integrated gradients) or Layer-wise Relevance Propagation (LRP), especially since there are connections between these techniques and Shapley values [1]."**
>
> In contrast to Neighbourhood SHAP, gradient-based methods require white-box access to the black box model as they need to compute the gradients. This requirement falls outside the scope of this paper which is why we did not include such a comparison in the first version of the paper.
>
> Since we appreciate the suggestion of the reviewer, we have now added a comparison with Integrated Gradients (IG) for the motivating example (Figure 1), and the MNIST experiment (Figure 7) in the appendix. IG, in particular, resembles feature-removal based approaches in so far as it requires baseline values to be specified. This baseline is typically defined to be just the mean of the feature. If a fixed baseline value is chosen, IG suffers under the same globality problem that was outlined for SHAP (Figure 1). Further, we see that the attributions of IG are quite similar to those of LIME in Figure 2. Figure 3 shows that masking with IG leads to the second-lowest average change in log odds and it thus underperforms (insignificantly) compared to the SHAP alternatives. A detailed comparison to Neighbourhood SHAP (i.e. black-box access vs white-box access, neighbourhood reference distribution vs baseline value) has been added to the Supplementary Material.
>
> [anonymous URL that we can share if needed]
> (Figure 1:) Feature attributions for motivating example.
>
> [anonymous URL that we can share if needed]
> (Figure 2:) Feature attributions on randomly picked MNIST images.
>
> [anonymous URL that we can share if needed]
> (Figure 3:) Change in log odds. See Appendix K.2 or [8] for a description of the experiment.
>
>
> **"The MNIST example is mostly anecdotal, stronger results would be quantitative. There is already a wave of works that are attempting to evaluate the quality of explanations [2-7]. Adopting some of these ideas/frameworks would give stronger evidence for claims."**
>
> We especially thank the reviewer for this suggestion, which came with a comprehensive list of references. Please note that fives types of quantitative evaluation methods are currently used in the paper.
> 1. In Appendix K.2, Figure 6 displays the change in log-odds obtained when masking sampled MNIST digits. This metric was adapted from [8] and as Yang and Kim [3] note, this kind of sensitivity measure is a useful evaluation metric.
> 2. We also computed the local prediction accuracy as an evaluation measure (see Figure 5), which is similarly also used as a quantitative evaluation measure by Plumb et al [1].
> 3. As a robustness metric, local Lipschitz estimates were computed on 100 test points on the MNIST dataset as proposed by [3] in Figure 8a in Appendix K.
> 4. Local Lipschitz estimates, as robustness metrics, on maximally perturbed MNIST images as proposed by [3] were included in Figure 8b in Appendix K.
> 5. Robustness to OOD detection is displayed in Figure 4a.
>
> Since our results can be applied to all types of tasks and models, we chose not to evaluate our methods using the metrics introduced in [5] and [3] as they are specific to image classification, nor the metrics from [2] and [4] which are meant for GNNs and RNNs respectively.
>
> Having said this, one of the suggested references that seems suitable for our setting is the ROAR framework by Hooker et. al [6]. For each observation, 20% of the most important features (measured by the magnitude of the absolute feature attributions) are imputed by their mean and the model is retrained. We distinguish between two settings: imputation with the (global) mean of the features and imputation with the local mean (i.e. the weighted mean of each feature where the weights were computed based on a distance metric to the observation).
>
> As we see in the table below, Neighbourhood and Smoothed SHAP result in a higher absolute change in test performance than Marginal SHAP on the boston housing data set. These are preliminary results and further experiments (additional runs for error bars and additional data sets) are currently running. Results will be added to the appendix.
>
> |   | Absolute change in test $R^2$ when imputing with local mean| Absolute change in Test $R^2$ when imputing with (global) mean
> --- | --- | ---
> |SHAP|0.349|0.154|
> |Neighbourhood SHAP|0.389|0.320|
> |Smoothed SHAP|0.177|0.399|
>
> **”Authors should write their Societal Impact statement.”**
>
> A paragraph titled ”Limitations and Societal Impact” has been added to the conclusion where we summarise the
> drawbacks of Neighbourhood SHAP that were addressed throughout the paper.
>
> ### References (references 2-6 match the reviewer’s referencing)
> [1] Plumb, Gregory, Denali Molitor, and Ameet Talwalkar. "Model agnostic supervised local explanations." arXiv preprint arXiv:1807.02910 (2018).
>
> [2] Wiltschko AB, Sanchez-Lengeling B, Lee B, Reif E, Wei J, McCloskey KJ, Colwell L, Qian W, Wang Y. Evaluating Attribution for Graph Neural Networks.
>
> [3] Yang M, Kim B. Benchmarking attribution methods with relative feature importance. arXiv preprint arXiv:1907.09701. 2019 Jul 23.
>
> [4] Arras L, Osman A, Müller KR, Samek W. Evaluating recurrent neural network explanations. arXiv preprint arXiv:1904.11829. 2019 Apr 26.
>
> [5] Arras L, Osman A, Samek W. Ground truth evaluation of neural network explanations with CLEVR-XAI. arXiv preprint arXiv:2003.07258. 2020 Mar 16.
>
> [6] Hooker S, Erhan D, Kindermans PJ, Kim B. A benchmark for interpretability methods in deep neural networks. arXiv preprint arXiv:1806.10758. 2018 Jun 28.
>
> [7] Covert, I., Lundberg, S., and Lee, S.-I. (2020). Understanding global feature contributions with 344 additive importance measures. Advances in Neural Information Processing Systems, 33.
>
> [8] Lundberg, S. and Lee, S.-I. (2017). A unified approach to interpreting model predictions. arXiv preprint arXiv:1705.07874.

---

### Official Review · Reviewer_FFcC · 2021-07-16

**Rating:** 7
**Confidence:** 4

**Summary:**

The paper addresses an important problem in model-agnostic local explanation methods, i.e., using global distribution to simulate feature attributions for explaining local model behaviors. To this extent, the authors consider the formulation of neighborhood reference distributions that improve the local interpretability of Shapley values. Further, empirical evaluations show that Neighborhood Shapley values identify relevant feature attributions and improve on-manifold explainability and adversarial robustness against adversarial attacks.

**Limitations And Societal Impact:**

Yes

**Main Review:**

Strengths
1. The paper provides a theoretical understanding of attribution-based explanation methods for analyzing local model behavior.
2. The paper is well-written with clear notations and motivations for NeighborhoodShap and SmoothedShap.
3. The paper shows that how smoothing Shapley values can identify unstable feature attributions and improve output explanations.

Open questions
1. One of the underlying assumptions to generate neighborhood samples using either the underlying data distribution process or from an estimator that approximates that distribution. In a black-box setting, we do not have any knowledge of the training data distribution, and for the estimator, there would be an error in approximating the underlying distribution. It would be great if the authors can comment on this.

2. How practical is the assumption that "models can detect the true data manifold?"

3. In the image classification results, Neighbourhood DeepSHAP looks at images in the neighborhood (i.e., images with similar stroke) which can be misleading. For instance, in Fig. 7, the neighborhood stroke for the top-left stroke of digit '8' can be similar to strokes of other digits (e.g., '2', '6', and '9').

4. The proposed methods are extensively evaluated using diverse metrics such as adversarial robustness, prediction accuracy, and visual inspections. It would be great if the authors can comment why they did not use more established metrics from the literature like deletion/insertion metric?

**Time Spent Reviewing:**

10

---

> ### Author Response · Authors · 2021-08-10
> **We thank Reviewer FFcC for appreciating the importance of our work. Open questions have been addressed (here and in the paper), and additional quantitative evaluation metrics have been added.**
>
> 1. **"One of the underlying assumptions to generate neighborhood samples using either the underlying data distribution process or from an estimator that approximates that distribution. In a black-box setting, we do not have any knowledge of the training data distribution, and for the estimator, there would be an error in approximating the underlying distribution. It would be great if the authors can comment on this."**
>
> We assume that we have a reference dataset available, which is a common assumption in all papers that use SHAP, MAPLE [5], or the tabular version of LIME. Any removal-based method that does not require such access, will end up evaluating the black-box model off the data manifold which bears serious consequences as explained in the paper (ll. 125-129). We added this specification to the introduction of the paper.
>
> When using a density estimator for the generation of neighbourhood examples, there will indeed be an estimation or model misspecification error which is why we proposed the importance sampling approach instead, which is consistent and thus asymptotically unbiased.
>
> 2. **"How practical is the assumption that "models can detect the true data manifold?""**
>
> When training a classifier to distinguish between true and fake data samples, it learns to distinguish the data manifold and the manifold of the fake data on convergence. This is a common assumption, which is for instance the reason for the huge success of GANs [6].
>
> 3. **"In the image classification results, Neighbourhood DeepSHAP looks at images in the neighborhood (i.e., images with similar stroke) which can be misleading. For instance, in Fig. 7, the neighborhood stroke for the top-left stroke of digit '8' can be similar to strokes of other digits (e.g., '2', '6', and '9')."**
>
> In the image classification results, we only look at images labeled with the digits 3 or 8 (l. 283). A similar experimental setup was used by  Lundberg and Lee [4]. As a result, we know that any pixel attributions can only compare the underlying image to images with a digit 3 or an 8.
>
> 4. **"The proposed methods are extensively evaluated using diverse metrics such as adversarial robustness, prediction accuracy, and visual inspections. It would be great if the authors can comment why they did not use more established metrics from the literature like deletion/insertion metric?"**
>
> We thank Reviewer FFcC, especially for this suggestion. The submitted version of the paper contained the following evaluation metrics:
> * Local prediction MSE, adapted from [5], for performance measurement in Figure 5
> * Robustness to OOD detection in Figure 4a
> * Change in log-odds when masking with SHAP values as proposed by [4] in Figure 7b in Appendix K
> * Local Lipschitz estimates as a robustness metric computed on 100 test points on the MNIST dataset as proposed by [3] in Figure 8a in Appendix K
> * Local Lipschitz estimates on maximally perturbed MNIST images as proposed by [3] in Figure 8b in Appendix K
>
> The third evaluation metric listed is indeed one kind of deletion metric. We further note that deletion/insertion metrics have traditionally been used on image datasets and are not established to evaluate the performance of explainability tools on tabular data sets, which is the focus of our work, as far as we are concerned, see e.g. [1,4,7,8].
>
> As we appreciate the suggestion of the reviewer, we added an evaluation metric taken from the ROAR framework developed by Hooker et. al [2], which is a recently proposed deletion metric. For each observation, 20% of the most important features (measured by the magnitude of the absolute feature attributions) are imputed by their mean and the model is retrained. Eventually, we compare the resulting changes in predictive accuracy. We distinguish between two settings: imputation with the (global) mean of the features and imputation with the local mean (i.e. the weighted mean of each feature where the weights were computed based on a distance metric to the observation).
>
> As we see in the table below, Neighbourhood and Smoothed SHAP result in a higher absolute change in test performance than Marginal SHAP on the boston housing data set. These are preliminary results and further experiments (additional runs for error bars and additional data sets) are currently running. Results will be added to the appendix.
>
> |   | Absolute change in test $R^2$ when imputing with local mean| Absolute change in Test $R^2$ when imputing with (global) mean
> --- | --- | ---
> |SHAP|0.349|0.154|
> |Neighbourhood SHAP|0.389|0.320|
> |Smoothed SHAP|0.177|0.399|
>
>
> ### References
>
> [1] Merrick L, Taly A. The explanation game: Explaining machine learning models using shapley values. In International Cross-Domain Conference for Machine Learning and Knowledge Extraction 2020 Aug 25 (pp. 17-38). Springer, Cham.
>
> [2] Hooker S, Erhan D, Kindermans PJ, Kim B. A benchmark for interpretability methods in deep neural networks. arXiv preprint arXiv:1806.10758. 2018 Jun 28.
>
> [3] Alvarez-Melis, D. and Jaakkola, T. S. (2018). On the robustness of interpretability methods. arXiv preprint arXiv:1806.08049.
>
> [4] Lundberg, S. and Lee, S.-I. (2017). A unified approach to interpreting model predictions. arXiv preprint arXiv:1705.07874.
>
> [5] Plumb, Gregory, Denali Molitor, and Ameet Talwalkar. "Model agnostic supervised local explanations." arXiv preprint arXiv:1807.02910 (2018).
>
> [6] Goodfellow, I., Pouget-Abadie, J., Mirza, M., Xu, B., Warde-Farley, D., Ozair, S., ... & Bengio, Y. (2014). Generative adversarial nets. Advances in neural information processing systems, 27.
>
> [7] Bhargava, V., Couceiro, M., and Napoli, A. (2020). Limeout: An ensemble approach to improve process fairness. arXiv preprint arXiv:2006.10531.
>
> [8] Botari, T., Hvilshøj, F., Izbicki, R., and de Carvalho, A. C. (2020). Melime: Meaningful local explanation for machine learning models. arXiv preprint arXiv:2009.05818.

---

> > ### Comment · Reviewer_FFcC · 2021-08-13
> > **Discussion response**
> >
> > I appreciate the authors for providing a detailed response to my concerns and running additional experiments. While I agree that the ROAR metric is based on deletion metric, but it is highly dependent on training the model multiple times using perturbed datasets. The metric choice is something that is not the scope of the work, so that's fine.
> >
> > **Re: 3 --** In that case, the results on MNIST are clearly on a manually crafted small subset of the original MNIST dataset. It is still unclear how this approach would scale to datasets with a higher number of classes, even if it's as simple as MNIST.

---

> > > ### Author Response · Authors · 2021-08-16
> > > **Full reference data set MNIST experiments added to the appendix**
> > >
> > > We thank Reviewer FFcC for acknowledging the efforts made in our previous comment.
> > >
> > > Regarding the ROAR metric, the retraining phase is indeed a limitation of this method. However, it does remove a substantial drawback of other deletion metrics where samples with a subset of features removed are drawn from a different distribution. The retraining phase in ROAR supports a common assumption in machine learning that the training and evaluation set are identically distributed (see Hooker et. al [1]).
> > >
> > > We note that the full MNIST data set is used to train the neural network and only the reference distribution was constrained. We followed the experimental setup of Lundberg and Lee [2] for the results, which was motivated in [2] by the easier interpretability of a restricted reference data set. With a more complicated reference data set it is unclear which alternative digit the attributions refer to, as noted by reviewers. Following Lundberg and Lee we made this choice in order to allow for an interpretable graphical representation of the performance of the compared XAI methods, noting that this does not limit our method. The number of digit classes used in the analysis does not affect the computational complexity, i.e. scalability, of the method. We thank the reviewer for raising this question and we will provide further comments in the main paper as well as a log-odds analysis for the full reference data set in the appendix.
> > >
> > >
> > > ### References
> > > [1] Hooker S, Erhan D, Kindermans PJ, Kim B. A benchmark for interpretability methods in deep neural networks. arXiv preprint arXiv:1806.10758. 2018 Jun 28.
> > >
> > > [2] Lundberg, S. and Lee, S.-I. (2017). A unified approach to interpreting model predictions. arXiv preprint arXiv:1705.07874.

---

### Official Review · Reviewer_RMYP · 2021-07-20

**Rating:** 7
**Confidence:** 4

**Summary:**

SHAP computes feature attribution by estimating the expected change in model prediction when a feature is included into a random subset of the rest of features. To account for the "removed features", conventional SHAP employs a "global reference distribution" (typically the marginal or conditional global population) from which values are sampled for the missing (removed) features. This paper argues that such global reference distribution can be too coarse-grained as to understand the local behavior of model prediction. To increase local fidelity, the paper proposes Neighborhood SHAP which considers a more local reference distribution to sample from, which encompasses conventional SHAP as a special instance. The core idea of Neighborhood SHAP is the use of local reference distribution, which depends on not only the original marginal (or conditional) reference distribution considered in conventional SHAP but also a distance-based distribution that gives higher weights to values around the example to be explained. Such design allows Neighborhood SHAP to capture more fine-grained local behavior but also improves on-manifold explainability. The paper then discusses Smoothed SHAP (averaging SHAP around a local neighborhood) as a closely related variant. Experimental results demonstrate that Neighborhood SHAP increases local fidelity and possesses higher robustness to adversarial attacks, and Smoothed SHAP shows difference between attributions on the test instance and those on neighboring observations.

**Limitations And Societal Impact:**

The authors have pointed out some limitations of the work and how they managed to address them in the Discussion section.

**Main Review:**

**Update**: Thank you to the authors for the well-thought responses, and conducting the suggested additional experiments. My questions have been addressed well, and I am happy to see the inclusion of additional discussions/experiments in the revision. I believe the paper cast an important notion on the "locality" of explanations, and has provided a comprehensive investigation on the topic.

Overall: The paper presents a clean framework to view current SHAP approaches, and the proposed Neighborhood SHAP is well-motivated. The formulation of Neighborhood SHAP naturally leads to a closely related variant, Smoothed SHAP. While the experimental section can be improved by more ablations and comparisons (see below), current presentation of the paper does provide insights on how Neighborhood/Smoothed SHAP capture different explanation characteristics compared to the original SHAP.

Strengths:
- First of all, the paper is clearly-written and the organization makes it easy to follow.
- The paper presents a comprehensive and clear overview of related work on SHAP. The overview provides a clean formulation on SHAP (line 62-63) and enables a clear viewpoint that current approaches to SHAP (marginal or conditional) are actually in a sense "global" reference distributions. By providing motivating examples illustrating the potential pitfalls of considering global reference distributions, the paper makes the problem of improving local fidelity well-posed.
- The proposed Neighborhood SHAP is well-motivated, and through importance sampling, it could be viewed as a generalization of the original SHAP. This makes Neighborhood SHAP a natural extension of SHAP.
- I appreciate that the authors have discussed how to choose the bandwidth for Neighborhood SHAP, as it could be critical to the success of Neighborhood SHAP.
- I appreciate that the authors have included discussion on Smoothed SHAP in the paper in addition to Neighborhood SHAP, as Smoothed SHAP appears to be a very closely related and natural variant to think of.
- Examples shown demonstrate the advantage of Neighborhood SHAP to the original SHAP: increased local fidelity and robustness to attacks. Interpretation on Neighborhood SHAP and Smoothed SHAP is also insightful.

Apart from the above, followings are some main question I have when reading the paper and suggestions on room to improve:
- In Figure 1, does decreasing the size of local neighborhood (e.g. N(0, 1) --> N(0, 0.1)) affect (improves) the results for LIME and SHAP? Would gradient-based methods be better at capturing this kind of local behavior? In general, how would Neighborhood SHAP compare to gradient-based methods like Integrated Gradient?
- In Figure 5, what is the exact metric (formal definition) used for the local prediction accuracy? How does the metric capture local fidelity better? How does Neighborhood SHAP compare to LIME on local fidelity (not shown in Figure 5)? How to explain why Neighborhood SHAP with small bandwidth can sometimes has better performance even when the neighborhood size of test data is large?
- I'd expect to see more discussion and comparison between NeighborhoodSHAP and Smoothed SHAP. Can SmoothedSHAP be formulated as a special instantiation of Neighborhood SHAP (or vice versa)? Empirically. how does Smoothed SHAP compare to NeighborhoodSHAP on robustness against attacks / on local fidelity / and on explanation stability (sensitivity)? These seem to be missing now, and they are important to understand the different characteristics between the two.
- Eq.2: Does Smoothed SHAP suffer from off-manifold problem?
- Figure 6, first column, NeighborSHAP and SmoothedSHAP has very different attribution (different sign), how to explain this discrepancy and which one to use?
- While the presented qualitative examples are illustrative, I believe more quantitative measurements on evaluating explanations (e.g., robustness to attacks or explanation sensitivity) could make the results much more convincing.
- Recent work has pointed out that explanations using reference distribution could subject to unintentional biases [1], does Neighborhood/Smoothed SHAP mitigate this problem?

Minor questions:
- In line 79-81, it is mentioned that the weighting of LIME does not ensure higher weights are assigned to model evaluations for observations closer to $x$. Why is this the case? Could the authors elaborate more on this?
- In line 85-87, why is SHAP not making any assumption on the form of $f$? SHAP does assume a linear additive model to fit the black-box model. Could the author discuss more on this?
- Equation between Line 104-105: why does $r(X|x)$ disappear in the last (rightmost) term?

[1] Evaluations and Methods for Explanation through Robustness Analysis. Hsieh et al. 2021.

**Time Spent Reviewing:**

15

---

> ### Author Response · Authors · 2021-08-10
> **Part 2**
>
> **"While the presented qualitative examples are illustrative, I believe more quantitative measurements on evaluating explanations (e.g., robustness to attacks or explanation sensitivity) could make the results much more convincing."**
>
> We especially thank Reviewer RMYP for this suggestion. The submitted version of the paper contained the following evaluation metrics:
> * Local prediction MSE, adapted from [2], for performance measurement in Figure 5
> * Robustness to OOD detection in Figure 4a
> * Change in log-odds when masking with SHAP values as proposed by [5] in Figure 7b in Appendix K
> * Local Lipschitz estimates as a robustness metric computed on 100 test points on the MNIST dataset as proposed by [6] in Figure 8a in Appendix K
> * Local Lipschitz estimates on maximally perturbed MNIST images as proposed by [6] in Figure 8b in Appendix K
>
> As we appreciate the suggestion of the reviewer, we added an evaluation metric taken from the ROAR framework developed by Hooker et. al [4], which is a recently proposed deletion metric. For each observation, 20% of the most important features (measured by the magnitude of the absolute feature attributions) are imputed by their mean and the model is retrained. Eventually, we compare the resulting changes in predictive accuracy. We distinguish between two settings: imputation with the (global) mean of the features and imputation with the local mean (i.e. the weighted mean of each feature where the weights were computed based on a distance metric to the observation).
>
> As we see in the table below, Neighbourhood and Smoothed SHAP result in a higher absolute change in test performance than Marginal SHAP on the boston housing data set. These are preliminary results and further experiments (additional runs for error bars and additional data sets) are currently running. Results will be added to the appendix.
>
> |   | Absolute change in test $R^2$ when imputing with local mean| Absolute change in Test $R^2$ when imputing with (global) mean
> --- | --- | ---
> |SHAP|0.349|0.154|
> |Neighbourhood SHAP|0.389|0.320|
> |Smoothed SHAP|0.177|0.399|
>
> **"Recent work has pointed out that explanations using reference distribution could be subject to unintentional biases [1], does Neighborhood/Smoothed SHAP mitigate this problem?"**
>
> We thank the reviewer for drawing our attention to this reference which highlights the importance of our contribution, and which we now include in the paper. Indeed modelling feature removal with neighbourhood distributions is a meaningful way to introduce less bias than with global distributions as the reference values are closer to the test instance and the data manifold.
>
>
> **Response to “Minor questions”:**
> * **"In line 79-81, it is mentioned that the weighting of LIME does not ensure higher weights are assigned to model evaluations for observations closer to $x$. Why is this the case? Could the authors elaborate more on this?"**
>
> The weighting of LIME depends on the sampled coalitions and not the reference values. Imagine you want to explain a test instance $x=(7,7,7)$. If you sample the coalition $S=(1,1,0)$ and impute with a reference value $x^*=(7,7,1000)$, the black box value at $(x_S, x^*_{\bar{S}})$ receives a higher weight than the black box value at $(x_{S}, x’_{\bar{S’}})$ for $S’=(1,0,0)$ and $x’=(7, 6.9, 6.9)$ in the computation of LIME as $S’$ has fewer zeros.
>
> * **"In line 85-87, why is SHAP not making any assumption on the form of $f$? SHAP does assume a linear additive model to fit the black-box model. Could the author discuss more on this?"**
>
> Linear approximation methods like MAPLE [2] assume that the black box behaves linearly in a small neighbourhood around the test instance as they fit a tangent line approximation. In contrast, SHAP does not make any claims of explaining the form of the black box. Instead, it is defined as a difference in expectations. No matter the parameterization of the black box, the difference in expectations is always a well-defined measure as long as the black box is finite everywhere. Please refer to Appendix A for a comparison of SHAP and tangent line approximation methods.
>
> * **"Equation between Line 104-105: why does r(X knowing x) disappear in the last (rightmost) term?"**
>
> The expectation in ll. 104- 105 is taken w.r.t. $r(X^*|x)$. We approximate it by the empirical mean, i.e. sampling observations $x^*$ from $r(X^*|x)$ and taking the average. This is why the distribution $r$ is implicitly encoded in the samples but does not appear explicitly in the formula.
>
> ### References
>
> [1] Hsieh, C. Y., Yeh, C. K., Liu, X., Ravikumar, P., Kim, S., Kumar, S., & Hsieh, C. J. (2020). Evaluations and methods for explanation through robustness analysis. arXiv preprint arXiv:2006.00442
>
> [2] Plumb, Gregory, Denali Molitor, and Ameet Talwalkar. "Model agnostic supervised local explanations." arXiv preprint arXiv:1807.02910 (2018).
>
> [3] Slack, D., Hilgard, S., Jia, E., Singh, S., & Lakkaraju, H. (2020, February). Fooling lime and shap: Adversarial attacks on post hoc explanation methods. In Proceedings of the AAAI/ACM Conference on AI, Ethics, and Society (pp. 180-186).
>
> [4] Hooker S, Erhan D, Kindermans PJ, Kim B. A benchmark for interpretability methods in deep neural networks. arXiv preprint arXiv:1806.10758. 2018 Jun 28.
>
> [5] Lundberg, S. and Lee, S.-I. (2017). A unified approach to interpreting model predictions. arXiv preprint arXiv:1705.07874.
>
> [6] Alvarez-Melis, D. and Jaakkola, T. S. (2018). On the robustness of interpretability methods. arXiv preprint arXiv:1806.08049.

---

> ### Author Response · Authors · 2021-08-10
> **Part 1:  We thank Reviewer RMYP for the encouraging comments and suggestions made. A comparison to Integrated Gradients and additional evaluation metrics have been added. Open questions have now been addressed in the paper.**
>
> **"In Figure 1, does decreasing the size of local neighborhoods (e.g. N(0, 1) --> N(0, 0.1)) affect the results for LIME and SHAP?"**
>
> Please refer to Figure 2 in Appendix A for an illustration of the feature attributions of LIME and Neighbourhood SHAP for varying neighbourhood sizes. We observe that the variance of LIME’s attributions is higher than those of Neighbourhood SHAP for really small neighbourhood sizes (i.e. $\sigma=0.1$). Up to a neighbourhood size of 0.3, the positive attribution of Feature-2 for small negative $x_1$ remains significantly positive. In contrast, the change in the attributions for Neighbourhood SHAP with decreasing neighbourhood size is more stable and remain intuitive.
>
> **"How would Neighborhood SHAP compare to gradient-based methods like Integrated Gradient?"**
>
> In contrast to Neighbourhood SHAP, gradient-based methods require white-box access to the black box model as they need to compute the gradients. This requirement falls outside the scope of this paper which is why we did not include such a comparison in the first version of the paper.
>
> Since we appreciate the suggestion of the reviewer, we have now added a comparison with Integrated Gradients (IG) for the motivating example (Figure 1), and the MNIST experiment (Figure 7) in the appendix. IG, in particular, resembles feature-removal based approaches in so far as it requires baseline values to be specified. This baseline is typically defined to be just the mean of the feature. If a fixed baseline value is chosen, IG suffers under the same globality problem that was outlined for SHAP (Figure 1). Further, we see that the attributions of IG are quite similar to those of LIME in Figure 2. Figure 3 shows that masking with IG leads to the second-lowest average change in log odds and it thus underperforms (insignificantly) compared to the SHAP alternatives. A detailed comparison to Neighbourhood SHAP (i.e. black-box access vs white-box access, neighbourhood reference distribution vs baseline value) has been added to the Supplementary Material.
>
> [anonymous URL that we can share if needed]
> (Figure 1:) Feature attributions for motivating example.
>
> [anonymous URL that we can share if needed]
> (Figure 2:) Feature attributions on randomly picked MNIST images.
>
> [anonymous URL that we can share if needed]
> (Figure 3:) Change in log odds. See Appendix K.2 or [5] for a description of the experiment.
>
>
>
> **"In Figure 5, what is the exact metric (formal definition) used for the local prediction accuracy? How does the metric capture local fidelity better?"**
>
> As explained in lines 235 ff., the local prediction MSE (we replaced the term *accuracy* with *MSE*) is estimated by
> 1. Computing the SHAP values for all test instances $x$
> 2. Perturbing the test instances by sampling coalitions and imputing the removed features with reference values $x^*$
> 3. Computing the weighted MSE of the binary feature model $g_x(S)$ for predicting the black box at the perturbed test instances where the weights are given based on the exponential kernel of $(x-x^*)^2$
>
> The corresponding formula is
> $$\frac{1}{n_{test}}\sum_{x\in D_{test}}\frac{1}{\sum_{l=1}^L d(x, x_l^*)}d(x, x_l^*)\left(g_x(S_l)-f(x_{S_l}, x^*_{\bar{S_l}})\right)^2$$
> where $n_{test}$ denotes the size of the test set $D_{test}$ and $L$ the number of sampled coalitions. We chose $L=n_{test}$ in our experiments. Local fidelity is captured better than in the standard MSE metric by choosing small bandwidths for the exponential kernel $d$ (l. 238). We have now added this formula to the experimental section of the paper.
>
> **"How does Neighborhood SHAP compare to LIME on local fidelity (not shown in Figure 5)?"**
>
> The figures below show preliminary results for the MSE of TabularLIME, chosen since, as SHAP, it can also be expressed as a binary prediction model (Appendix E). Final results will be included in the appendix.
>
> [anonymous URL that we can share if needed]
> (Figure 1: boston)
>
> [anonymous URL that we can share if needed]
> (Figure 2: bike)
>
> Comparing these to the results presented in the paper corresponding to Neighbourhood SHAP, we see that the ranges differ, with TabularLIME's MSE always being larger than that of the SHAP alternatives: for the Boston data, the MSE for Neighbourhood SHAP ranges from 0 to 25, while for TabularLIME it starts at 10 and goes to 40. Similarly, on the bike data,   Neighbourhood SHAP performs significantly better than LIME (y-axis ranges from 0 to 5000 compared to 0 to 8000).
>
>
> **"How to explain why Neighborhood SHAP with small bandwidth sometimes has better performance even when the neighborhood size of test data is large?"**
>
> We also find this behaviour surprising. A plausible explanation is that, given the simplicity of the iris data, XGBoost models behave linearly in sufficiently small neighbourhoods,  resulting in Neighbourhood SHAP with small neighbourhood sizes outperforming those attributions with larger neighbourhood sizes.
>
> **"I'd expect to see more discussion and comparison between NeighborhoodSHAP and Smoothed SHAP. Can SmoothedSHAP be formulated as a special instantiation of Neighborhood SHAP (or vice versa)?"**
>
> As the reviewer noted, Neighbourhood SHAP is a standalone method whereas SmoothedSHAP was included as it naturally represents the complement of Neighbourhood SHAP: While Neighbourhood SHAP models feature removal with neighbourhood distributions, Smoothed SHAP models feature inclusion with neighbourhood distributions. Neighbourhood SHAP reflects feature attributions when contrasting with model behaviour for similar observations. Simplified, Neighbourhood SHAP answers the question, “Why did I get this model outcome compared to my peers?”. SmoothedSHAP summarises marginal Shapley values (which contrast against the entire population) within a neighbourhood, instead of at a single instance. Simplified, Smoothed SHAP answers the question, “Can I summarise the model outcome across me and my peers?”. This distinction will be included in a revised version of the paper.
>
> **"How does Smoothed SHAP compare to NeighborhoodSHAP on robustness against attacks / on local fidelity / and on explanation stability (sensitivity)?"**
>
> Neighbourhood SHAP focuses SHAP on greater locality; Smoothed SHAP summarises SHAP with less locality. Therefore Smoothed SHAP is less locally faithful than Neighbourhood SHAP, but its explanations are more stable (through averaging). As Smoothed SHAP averages over close marginal SHAP values, the imputed values do not lie on the data manifold, and the adversary can thus still detect off-manifold observations.
>
> However, the attack proposed by [3] is unstable since it relies on an OOD detector. As not all close-by imputed values are detected as being out of distribution, we indeed observe a non-significant increase in robustness against adversarial attacks for Smoothed SHAP with $\sigma_{smthd}=1$ as well: mean MSE$\pm$std of the relative attribution of race is $0.78\pm0.03$ for Smoothed SHAP, $0.81\pm0.03$ for SHAP, and $0.54\pm0.02$ for Neighbourhood SHAP.
>
> **"Eq.2: Does Smoothed SHAP suffer from off-manifold problem?"**
>
> Yes, as Smoothed SHAP is directly constructed from marginal Shapley values, it suffers from an off-manifold problem, just as marginal Shapley values do.
>
> **"Figure 6, first column, NeighborSHAP and SmoothedSHAP has very different attribution (different sign), how to explain this discrepancy and which one to use?"**
>
> For the adult data, we have focused on an individual for whom the model performs poorly. What makes this individual particular is his/her education number being 12, which corresponds to being “Associate-acdm”. Additionally, the relationship status and marital status of this individual (namely “Not-in-family” and “Never married”) are rare at age 43. Marginal SHAP gives a positive attribution to age. This means that overall, growing older increases the probability of one having a high income. However, the same feature attribution from Smoothed SHAP gets negative for large neighbourhood sizes. This shows that for people similar to our individual of interest (middle-aged, not in a family, highly educated), growing older decreases the chances of having a high income, potentially because they tend to be at their career’s peak. As for education, marginal SHAP gives a zero attribution to education which means it has no impact on income overall. SmoothedSHAP gives a positive attribution to education, showing that for the highly educated peers of this individual, education often played a big part in them having a high income. Neighbourhood SHAP gives a negative attribution to education which shows that the Academic we are considering would have had a higher expected salary if he had taken a different educational path. This observation is coherent with the fact that middle-aged people (age between 40 and 46) with an education number different to 12 are on average more likely to have a high income compared to the same age group with a different education number ($0.3639\pm0.005$ vs. $0.3333\pm0.000$).
> Overall, the two sign flips show how important it is not to apply marginal Shapley values blindly, but to plot Neighbourhood SHAP and Smoothed SHAP values of the normalised features over a range of bandwidths, as a powerful diagnostic and information tool.

---

### Decision · Program_Chairs · 2021-09-28

**Decision:**

Accept (Poster)

**Comment:**

I thank the authors for a good back-and-forth discussion during the review process.  All reviewers recommend accepting this paper, appreciating the well-motivated method, the extensive complete original work, and the valuable the theoretical insights.  I encourage the authors to integrate all the suggestions from the reviewers into the camera ready.

**Consistency Experiment:**

NeurIPS has a long history of experimentation. In 2014, NeurIPS ran an experiment in which 10% of submissions were reviewed by two independent committees to quantify the randomness in the review process. This year, we repeated a variant of this experiment to see how the quality of the review process has changed over time.  This paper was part of the experiment and was therefore assigned to two committees (consisting of reviewers, an Area Chair, and a Senior Area Chair) that reached independent decisions.  If both committees made the same recommendation, this recommendation was followed. If a single committee recommended acceptance, the paper was accepted (with the exception of a few cases in which the other committee identified what we considered a fatal flaw, e.g., an error in a key result).

This copy’s committee reached the following decision: **Accept (Poster)**

The other committee assigned to the paper recommended **Reject**.  You can find the other set of reviews, along with any follow up discussion with the authors here:
https://openreview.net/forum?id=UKoV0-BamX4